# From chemoproteomic-detected amino acids to genomic coordinates: insights into precise multi-omic data integration

Maria F Palafox[1,2,3] (ID), Heta S Desai[2,4], Valerie A Arboleda[1,3,4,5,6,*] (ID) & Keriann M Backus[2,4,5,6,7,8,**] (ID)

## Abstract

The integration of proteomic, transcriptomic, and genetic variant annotation data will improve our understanding of genotype–phenotype associations. Due, in part, to challenges associated with accurate inter-database mapping, such multi-omic studies have not extended to chemoproteomics, a method that measures the intrinsic reactivity and potential "druggability" of nucleophilic amino acid side chains. Here, we evaluated mapping approaches to match chemoproteomic-detected cysteine and lysine residues with their genetic coordinates. Our analysis revealed that database update cycles and reliance on stable identifiers can lead to pervasive misidentification of labeled residues. Enabled by this examination of mapping strategies, we then integrated our chemoproteomics data with computational methods for predicting genetic variant pathogenicity, which revealed that codons of highly reactive cysteines are enriched for genetic variants that are predicted to be more deleterious and allowed us to identify and functionally characterize a new damaging residue in the cysteine protease caspase-8. Our study provides a roadmap for more precise inter-database mapping and points to untapped opportunities to improve the predictive power of pathogenicity scores and to advance prioritization of putative druggable sites.

**Keywords** amino acid reactivity; chemoproteomics; genetic pathogenicity prediction; inter-database mapping; multi-omics

**Subject Category** Proteomics

**Mol Syst Biol. (2021) 17: e9840**

## Introduction

Understanding how proteins work is the bedrock of functional biology and drug development. The identification of amino acids that directly regulate a protein's activity (e.g., catalytic residues, residues that drive interactions, or residues important for folding or stability) is an essential step to functionally characterize a protein. Delineation of amino acid-specific functions is typically accomplished using site-directed mutagenesis (Hemsley *et al*, 1989; Starita *et al*, 2015). While such studies can identify functional hotspots in human proteins, they are typically limited in scope and largely restricted to proteins easily expressed *in vitro*. With the advent of next-generation sequencing and CRISPR-based mutagenesis, deep mutational analysis can now be scaled to individual genes (e.g., *TP53* and *BRCA1*) (Starita *et al*, 2015; Boettcher *et al* 2019), but such studies have not been extended genome-wide.

This problem of identifying the functional properties of a specific amino acid parallels one of the central challenges of modern genetics: interpreting the pathogenicity of the millions of genetic variants found in an individual's genome. Many computational methods, such as M-CAP (Jagadeesh *et al*, 2016), Combined Annotation Dependent Depletion (CADD) (Kircher *et al*, 2014), PolyPhen (Adzhubei *et al*, 2010), and SIFT (Vaser *et al*, 2016) integrate data such as sequence conservation, metrics of sequence constraint, and other functional annotations to provide a quantitative assessment of variant deleteriousness. In the absence of experimental data, these scores provide a metric to rank genetic variants for their effect on a phenotype, something particularly important in the era of genome-wide association and sequencing studies.

Beyond genetic variation, a frequently overlooked parameter that defines functional hotspots in the proteome is amino acid side chain reactivity, which can fluctuate depending on the residue's local and 3-dimensional protein microenvironment. Mass spectrometry-based chemoproteomics methods have been developed that can assay the

1  Department of Human Genetics, David Geffen School of Medicine, UCLA, Los Angeles, CA, USA
2  Department of Biological Chemistry, David Geffen School of Medicine, UCLA, Los Angeles, CA, USA
3  Department of Pathology and Laboratory Medicine, David Geffen School of Medicine, UCLA, Los Angeles, CA, USA
4  Molecular Biology Institute, UCLA, Los Angeles, CA, USA
5  Jonsson Comprehensive Cancer Center, UCLA, Los Angeles, CA, USA
6  Eli and Edythe Broad Center of Regenerative Medicine and Stem Cell Research, UCLA, Los Angeles, CA, USA
7  Department of Chemistry and Biochemistry, College of Arts and Sciences, UCLA, Los Angeles, CA, USA
8  DOE Institute for Genomics and Proteomics, UCLA, Los Angeles, CA, USA
   *Corresponding author: Tel: +1 310 983 3358; E-mail: varboleda@mednet.ucla.edu
   **Corresponding author: Tel: +1 310 206 8617; E-mail: kbackus@mednet.ucla.edu

intrinsic reactivity of thousands of amino acid side chains in native biological systems (Weerapana *et al*, 2010; Backus *et al*, 2016; Hacker *et al*, 2017). Using these methods, previous studies, including our own, revealed that "hyper-reactive" or pKa-perturbed cysteine and lysine residues are enriched in functional pockets. These chemoproteomics methods can even be extended to measure the targetability or "druggability" of amino acid side chains, which has revealed that a surprising number of cysteine and lysine side chains can also be irreversibly labeled by small drug-like molecules (Weerapana *et al*, 2010; Backus *et al*, 2016; Hacker *et al*, 2017). Complicating matters, for the vast majority of these chemoproteomic-detected amino acids (CpDAA), the functional impact of a missense mutation or chemical labeling remains unknown. Integrating chemoproteomics data with genomic-based annotations represents an attractive approach to stratify CpDAA functionality and to identify therapeutically relevant disease-associated pockets in human proteins.

Such multi-omic studies require mapping a protein's sequence back to genomic coordinates, through the transcript isoforms, in essence reverse engineering the central dogma of molecular biology. Accurate mapping between amino acid positions and genomic coordinates remains particularly challenging, due in part to the diversity of cell type-specific transcript and protein isoforms and the non-linear relationship between gene, transcript, and protein sequences. One approach to address these challenges is through proteogenomics (Ruggles *et al*, 2017), where custom FASTA files are generated from whole exome or RNA-sequencing data. However, such approaches are not scalable or cost-effective. Furthermore, many proteomic datasets, particularly previously acquired and public datasets, lack matched genomic data, precluding proteogenomic analysis.

Many computational tools have been developed for inter-database mapping, including using unique identifiers (Durinck *et al*, 2009; Smith *et al*, 2019; Agrawal & Prabakaran, 2020), methods to map genomic coordinates to protein sequences and structures (David & Yip, 2008; Sehnal *et al*, 2017; Sivley *et al*, 2018; Stephenson *et al*, 2019), and tools for codon-centric-based annotation of genetic variants (Gong *et al*, 2014; Schwartz *et al*, 2019). One key application of these tools is the improved prediction of variant pathogenicity (Guo *et al*, 2017). However, while many predictive genetic scores are built on the GRCh37 genome assembly (frozen in 2014), the UniProt Knowledge Base (UniProtKB) (McGarvey *et al*, 2019) proteomic reference is based on genome assembly GRCh38. Further complicating data integration, the unsynchronized and frequent updates to widely used databases, such as UniProtKB and Ensembl, result in a constantly evolving landscape of genome-, transcriptome- and proteome-level sequences and annotations, which further confounds multi-omic data integration, particularly for residue-level analyses.

Focusing initially on previously identified CpDAAs (Weerapana *et al*, 2010; Backus *et al*, 2016; Hacker *et al*, 2017), we first assess how choice of databases, including release dates, and the use of isoform-specific, versioned or stable identifiers impact residue-coordinate mapping and the fidelity of data integration. We then apply an optimized mapping strategy to annotate CpDAA positions with predictions of genetic variant pathogenicity, for both previously published and newly generated chemoproteomic analyses of amino acid reactivity. Our study uncovers key sources of inaccurate

mapping and provides fundamental guidelines for multi-omic data integration. We also reveal that highly reactive cysteines, including those identified previously (Weerapana *et al*, 2010) and newly identified CpDAAs, are enriched for genetic variants that have high predicted pathogenicity (high deleteriousness), which supports both the utility of predictive scores to further power proteomics datasets and the use of chemoproteomics to add another layer of interpretation to missense genetic variants. As many databases move to GRCh38, we anticipate that our findings will provide a roadmap for more precise inter-database comparisons, which will have wide-ranging applications for both the proteomics and genetics communities.

# Results

## Characterizing the dynamic mapping landscape relevant to CpDAA data integration

Our first step to achieve high-fidelity multi-omic data integration was to establish a comprehensive set of test data. For this, we aggregated publicly available cysteine and lysine chemoproteomics datasets (Weerapana *et al*, 2010; Backus *et al*, 2016; Hacker *et al*, 2017), resulting in a total of 6,510 CpD cysteines and 9,327 CpD lysines detected in 4,119 unique proteins. These 15,837 CpDAAs are further sub-categorized by the residues labeled by cysteine- or lysine-reactive probes (iodoacetamide alkyne [IAA] or pentynoic acid sulfotetrafluorophenyl ester [STP], respectively) and those residues with additional measures of intrinsic reactivity (categorized as high-, medium-, and low-reactive residues; Dataset EV1).

As our overarching objective was to characterize CpDAAs using functional annotations based on different versions of protein, transcript, and DNA sequences (Fig 1A), our next step was to develop a high-fidelity data analysis pipeline for intra- and inter-database mapping. To guide our analyses, we first referenced established methods for such data mapping, including ID mapping (Huang *et al*, 2008; Meyer, Geske, & Yu, 2016; Xin *et al*, 2016), residue–residue mapping (Martin, 2005; David & Yip, 2008; Dana *et al*, 2019), and residue–codon mapping (Zhou *et al*, 2015; Li *et al*, 2016) (See Appendix Table S1 for detailed descriptions of each type of mapping).

We suspected that the frequent and unsynchronized update cycles of independent databases (Fig 1B; Dataset EV2) might complicate accurate residue-level mapping. Supporting this hypothesis, quantification of the average update cycle for each database across this time period revealed that UniProtKB has the shortest mean update cycle (~ 6 weeks; Fig 1C). In contrast, NCBI is only updated yearly. These different update cycles can create a lag between versions of databases used to create identifier cross-reference (a.k.a. External Reference [xref]) files (Appendix Table S1). For example, ID mapping files provided by Ensembl for UniProtKB proteins may not share identical sequences if not used within the short 4-week window between UniProtKB updates.

To enable further characterization of how database update cycles and mapping strategy impact the fidelity of data integration, we collected a test set of Ensembl releases (Appendix Fig S1 and Dataset EV3). Specific releases were prioritized that (i) represented reference releases based on the GRCh37 or GRCh38 reference

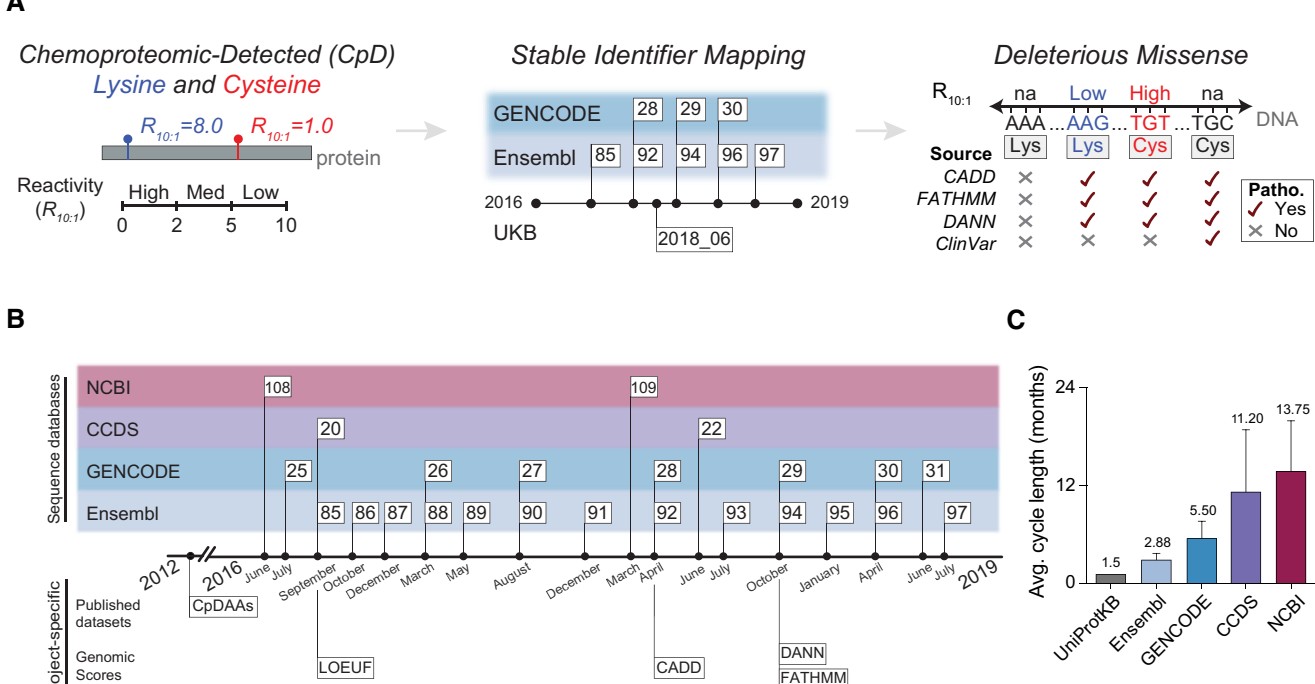

**Figure 1. Landscape of sequence annotation information updates.**

A   Schematic representation of mapping chemoproteomic-detected amino acids (CpDAAs) to pathogenicity scores.

B   Timeline of gene annotation database release dates and project-specific datasets, including Ensembl releases tested for compatibility (Fig 2) to CpDAA coordinates based on canonical UniProtKB protein sequences and the database reference corresponding to the genomic pathogenicity scores (Fig 3).

C   Average database release cycle length for releases between August 2013 and July 2019. All values are mean ± SD. Total of 25 Ensembl, 13 GENCODE, six CCDS (homo sapien only), and five NCBI releases were counted. UniProtKB value was calculated by taking the average of release cycle lengths reported on the UniProt website.

genome, (ii) were compatible with the latest Consensus Coding Sequence (CCDS) update for the human genome (release 22), (iii) were used in database for nonsynonymous functional predictions (dbNSFP) v4.0a and CADDv1.4, two resources that integrate functional annotations for all possible nonsynonymous single nucleotide variants (SNV) (Kircher *et al*, 2014; Liu *et al*, 2016; Rentzsch *et al*, 2019), and (iv) were associated with a commonly used version of the Ensembl Variant Effect Predictor (VEP) (McLaren *et al*, 2016).

With these prioritized datasets in hand, we next tracked the loss of CpDAA-containing protein IDs during intra-database mapping of UniProtKB releases and inter-database ID mapping to different Ensembl releases. Gratifyingly, only a handful of the original 4,119 protein IDs were lost due to database updates, both for Ensembl (e.g., 37 IDs for v97 release of Ensembl) and for UniProtKB (e.g., 26 IDs for 2012 UniProtKB; Appendix Fig S1, Datasets EV1 and EV4). The greatest identifier loss was observed from mapping UniProtKB-based legacy data to the 2018 UniProtKB-SwissProt CCDS cross-referenced curation of the human proteome, with 119 IDs not found in the 2018 dataset. We ascribe this identifier loss to both UniProtKB updates and to the higher level of curation for proteins in the 2018 dataset, which includes only Swiss-Prot canonical protein sequences with a cross-referenced ("xref") entry term in the CCDS database. Of note, CCDS gene IDs are manually reviewed and linked to UniProtKB-SwissProt. The TREMBL database is comprised of automatically generated protein IDs, which, as a result, comprises a substantially larger set of UniProtKB IDs, when compared to the manually curated SwissProt CCDS subset (Appendix Fig S2). From these analyses, we concluded that using the CCDS UniProtKB release was optimal for integrating functional annotations with chemoproteomics datasets.

### Updates to canonical sequences assigned to UniProtKB stable identifiers can lead to intra-database mismapping of CpDAAs

Proteomics datasets, including published CpDAA datasets, are routinely searched against FASTA files containing only canonical UniProtKB proteins (Appendix Table S1), for two main reasons. First, canonical proteins reduce the redundancy and complexity of proteome search databases. Second, these sequences are identified by stable identifiers (also known as the UniProtKB primary accessions) and offer the seeming advantage of remaining constant through database update cycles. However, one particularly confusing aspect of the stable identifier is that the word "stable" in this context does not mean permanent or immutable. Specifically, the associated sequence linked to a stable identifier can change over database releases.

Therefore, we next assessed whether and to what extent updates to the canonical sequences assigned to UniProtKB stable identifiers resulted in mismapping. To confirm the integrity of our CpDAA dataset, we started this process by validating that over 99% of the

CpDAA protein IDs and residue positions matched with those found in a 2012 UniProt FASTA file, corresponding to the reference proteome originally used to process the datasets (see Materials and Methods and Dataset EV1). The small fraction of data lost was due to missing stable identifiers and mis-matched CpDAA positions, which likely stems from slight inconsistencies between the original processing pipeline and our current workflow. We then mapped the 6,404 CpD cysteines and 9,213 CpD lysines from 4,084 canonical proteins identified in the 2012 dataset to the 2018 UniProtKB CCDS canonical sequence subset of the human proteome. Mapping to CCDS sequences enabled us to take advantage of the extensive array of tools that facilitate forward and reverse annotation between gene, transcript, and protein sequences and would allow for residue-specific mapping to genomic functional annotations (Dataset EV5) (Zhou et al, 2015; Meyer, Geske, & Yu, 2016; McGarvey et al, 2019). Updating to the 2018 release was a requisite step for using these tools, as they overwhelmingly require recent cross-reference files using the newest reference genome GRCh38. For all CpDAA positions, we performed residue–residue mapping—defined as a one-to-one correspondence between amino acids in proteins from different databases or release dates—to match the 2012 canonical UniProtKB sequences with their 2018 counterparts (Dataset EV4). This dataset mapping resulted in the loss of 121 protein IDs, with 108 simply not found in the 2018 reference file and the remaining 13 found to have different canonical sequences, resulting in mismapping or loss of the originally identified CpDAA residues.

The high concordance between these two UniProtKB releases, separated by 6 years, indicates that for the vast majority of UniProtKB updates, differences in release date should not complicate re-mapping legacy proteomics data to more recently released gene, transcript, and protein sequences. However, we were surprised to find that several widely studied proteins, including protein arginine N-methyltransferase 1 (PRMT1 or ANM1, Q99873), serine/threonine protein kinase, (SIK3; Q9Y2K2) (Walkinshaw et al, 2013), and tropomyosin alpha-3 chain (TPM3, P06753), had canonical protein sequence differences resulting in all or nearly all CpDAA positions to be missed using the 2018 position index (Dataset EV4). We observed two main reasons for these losses: (i) changes to the canonical sequence associated with the UniProtKB stable ID and (ii) changes to which isoform is assigned as the canonical sequence. While both 2012 and 2018 sequences of PRMT1 are associated with UniProtKB stable ID Q99873, the 2018 sequence contains an additional short N-terminal sequence, not present in the 2012 sequence (Fig 2A). As a result, all 13 PRMT1 CpDAAs failed to map to the 2018 UniProtKB release. In the 2012 release of UniProtKB, the canonical sequence of the peptidyl-prolyl cis-trans isomerase FKBP7 is associated with the versioned (isoform) ID Q9Y680-1, whereas in the 2018 release, the canonical sequence is associated with the versioned (isoform) ID Q9Y680-2, which lacks a short sequence (AAΔ125:162) in the middle of the protein. For FKBP7, this update fortuitously does not result in loss of CpD Lys83, as it is located N-terminal to the deletion. These updates to the protein sequence are, in essence, masked by the stable IDs, which do not flag sequence updates or changes to which isoform sequence is assigned as the canonical. Exemplifying this problem, we identified 45 stable identifiers with non-identical canonical protein sequences in the 2012 and 2018 UniProtKB releases (Dataset EV4).

To further understand how the presence or absence of protein isoforms impacts the fidelity of data mapping during intra-database (UniProtKB) mapping, we identified all isoforms associated with CpDAA stable protein IDs. Analysis of this dataset revealed that 58% of protein stable IDs have between 2–5 associated isoform sequences (Fig 2B). Catenin delta-1 protein (CTNND1, O60716) had 32 isoforms, which was the greatest number of isoforms in our dataset (Dataset EV6). Protein isoforms are identified by the "-X" after the UniProtKB ID, where X represents the isoform name. A common assumption of most mapping tools and proteomics databases is that the "-1" sequence is the canonical sequence. However, a key finding from our isoform analysis is that the canonical sequence does not always correspond to the "-1" isoform ID provided by UniProtKB. In fact, for 288 proteins in the UniProtKB 2018 release, the non-"-1" entry corresponds to the canonical isoforms, and for 55 CpDAA-containing proteins in our dataset (~ 2%), the canonical sequence is not the "-1" isoform (Fig 2C and Dataset EV7). Strikingly, the canonical sequence can even be the "-10" isoform, as is the case for the Ras-associated and pleckstrin homology domains-containing protein (RAPH1, Q70E73). In the context of database mapping, all of these non-"-1" canonical proteins will likely result in mismapping using established tools.

## Accurate residue-level inter-database mapping between UniProtKB and Ensembl is dependent on database update cycles

To investigate how sequence versions impact inter-database mapping, we next turned to ID cross-reference files (Dataset EV3) that are released by Ensembl and UniProtKB. Cross-reference files can be used to convert between UniProtKB and Ensembl ID types. Three major challenges arise with ID cross-referencing: (i) when cross-reference stable IDs match, but corresponding sequences are not identical, (ii) multi-mapping, where a UniProtKB ID maps to many Ensembl protein (ENSP), transcript, and gene IDs, and (iii) when the origin, both the time of the releases and the specific database provided cross-reference files used, determines the mapping accuracy of datasets.

Glucose-6-phosphate dehydrogenase (G6PD, P11413) exemplifies how sequence updates associated with a stable ID can lead to mismapping of gene-, transcript-, and protein-level annotations for CpDAAs (Fig 2D). For G6PD, the same UniProtKB ID maps to four unique ENSP IDs with identical sequences (see first row in "Identical") as well as four different ENSP IDs with non-identical sequences (see second row in "Non-identical"). For G6PD, this significant redundancy is also observed at the gene and transcript level, both for stable and versioned IDs (Fig EV1A; Dataset EV8). Overall, genes undergo the highest frequency of sequence re-annotation due to continual refinement of the reference genome. In contrast, protein IDs remain largely fixed across releases (Fig EV1B; Dataset EV9).

To assess how pervasive multi-mapping is across the entire CpDAA dataset, we quantified the mean number of Ensembl IDs per UniProtKB ID. We counted both versioned and stable Ensembl IDs types (gene, transcript, and protein IDs), for all CpD UniProtKB proteins grouped by single (Fig EV1C) or multi-isoform (Fig EV1D; Dataset EV10) associated stable IDs. We suspected that database updates for all data types (gene, transcript, and protein) and the presence of UniProtKB isoforms would contribute to the observed

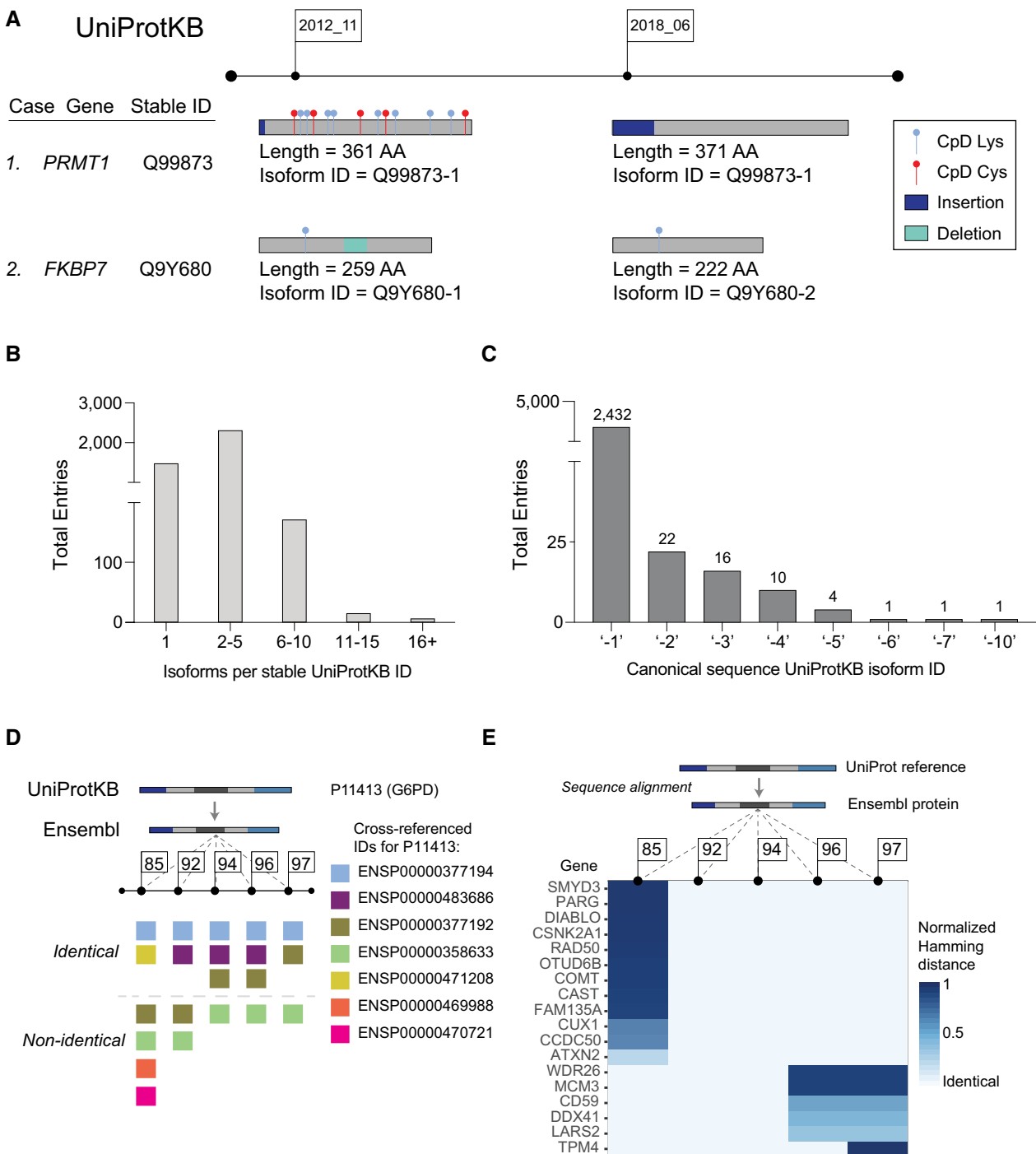

**Figure 2. Challenges with residue-level mapping and UniProtKB canonical protein sequences.**

A   Schematic depiction of mapping scenarios from updating chemoproteomic-detected protein sequences using stable or versioned identifiers.

B   Distribution of number of isoforms per stable UniprotKB ID for 3,953 detected proteins.

C   Frequency of specific isoform name for 2,487 multi-isoform UniProtKB canonical proteins.

D   Schematic depiction of glucose-6-phosphate dehydrogenase (G6PD, UniProtKB ID P11413) cross-referencing both identical and non-identical sequences of Ensembl stable IDs from five releases.

E   Heatmap of protein sequence distance scores for detected UniProtKB and cross-referenced Ensembl proteins from five releases. Each gene name corresponds to one unique stable Ensembl protein ID.

multi-mapping of CpD protein IDs in our dataset. Of note, Ensembl versioned IDs indicate changes to the associated sequence rather than the presence of isoforms. For example, for protein tropomyosin alpha-4 chain (TPM4, P67936), during the update from v96 to v97, the stable protein identifier showed version change from ".3" to ".4" (ENSP00000300933.3 to ENSP00000300933.4), which corresponds to a difference of 165 amino acids in the primary sequence caused by the update (Dataset EV11). Not surprisingly, we found that UniProtKB stable identifiers with multiple associated protein isoforms have a higher average of cross-referenced Ensembl ID types per UniProtKB stable identifier, when compared to UniProtKB stable IDs associated with only one protein isoform. In addition, single isoform UniProtKB stable IDs are more likely to cross-reference identical ENSPs, when compared to multi-isoform UniProtKB stable IDs (Appendix Figs S3 and S4).

One last challenge we identified is that the origin of the cross-reference file (whether it was created by UniProtKB or by Ensembl) affected the outcome of our mapping procedures. Across the five Ensembl releases, only 56.9% of all Ensembl-UniProtKB cross-referenced IDs had identical protein sequences (Appendix Fig S3; Dataset EV8). We then used a cross-reference file from UniProtKB that, unlike the Ensembl mapping files, contains mappings with canonical isoform protein identifiers for UniProtKB proteins to Ensembl stable protein IDs, to test whether inclusion of isoform name details improves the accuracy of inter-database ID mapping. This approach allowed for > 99% identical protein sequence cross-references for UniProtKB-ENSP IDs and substantially reduced the burden of identifier multi-mapping (Appendix Fig S4; Dataset EV12). Our study demonstrates that high-fidelity ID cross-referencing requires attention to details regarding database updates, multi-mapping, and identifier types used in cross-reference file sources. We also observed that sequences associated with mapped UniProtKB and Ensembl stable IDs varied significantly in alignment distance depending on the Ensembl version (Fig 2E; Appendix Fig S5; Dataset EV11), with temporally close releases showing generally greater sequence similarity.

## Assessment of pathogenicity predictions for CpD cysteine and lysine codons, using residue–codon mapping

Our next objective was to apply residue–codon mapping to the prioritization of functional CpDAAs. Cysteines and lysines are both highly conserved, with 97% (Miseta & Csutora, 2000) and 80% (Hacker et al, 2017) median conservation, respectively. Consequently, sequence motif conservation cannot distinguish between functional and non-functional residues within chemoproteomics datasets. To identify cysteine- and lysine-centric genetic features suitable for pathogenicity prioritization, we tailored our pipeline to reverse-translate CpD cysteine and lysine positions in canonical UniProtKB proteins to genomic coordinates from both major genome assemblies (GRCh37 and GRCh38) and genomic-based functional annotations. For all proteins within our CpDAA dataset, referred to as detected proteins, we also processed undetected equivalent residue types in CpD Cys- and/or CpD Lys-containing proteins (Fig EV2). Cysteines and lysines were required to have valid coordinates in GRCh37 and GRCh38 reference genome assemblies, as some functional genetic variant annotations are only available in one genome assembly (Dataset EV13). Probe-labeled cysteines and lysines represent ~ 15% of all cysteines (6,057 CpD Cys out of 40,107 total Cys) and ~ 6% of all lysines (8,868 CpD Lys out of 149,520 total Lys) found in chemoproteomic-identified proteins ($n$ = 3,840 UniProtKB IDs successfully mapped; Fig 3A and B).

Next, genomic coordinates of cysteine and lysine codons from 3,840 detected proteins were annotated by a panel of functional scores (Quang, Chen, & Xie, 2015; Shihab et al, 2015; Ioannidis et al, 2016; Jagadeesh et al, 2016; preprint: Samocha et al, 2017; Sundaram et al, 2018; Rentzsch et al, 2019). With the goal of assessing the correlation between individual scores and chemoproteomics identification labels, we selected complementary pan-genome and missense deleteriousness prediction scores (Dataset EV13) based on either GRCh37 or GRCh38 reference genome assemblies for our analysis. For the CADD score, which is available for both assemblies, we observed a trend of slightly higher scores with CADD38 compared to CADD37 (Appendix Fig S6). We calculated the Spearman's correlation of scores for all possible nonsynonymous SNVs overlapping cysteine and lysine codons and saw a higher correlation between the deleteriousness predictions for CpD cysteine substitutions (Fig 3C; Dataset EV14) than for CpD lysine substitutions (Fig 3D; Dataset EV14). For the subset of scores that provide deleteriousness scores for all possible nonsynonymous variants, we did not observe substantial differences between the correlation of scores for chemoproteomic-detected and -undetected lysines or cysteines (Appendix Fig S7; Dataset EV15).

Pathogenicity thresholds, which are provided by a subset of the scores investigated (e.g., CADD, functional analysis through hidden markov models [fathmm-MKL], and Deleterious Annotation of genetic variants using Neural Networks [DANN]), provide a useful cut-off for assessing whether substitutions at specific amino acids are likely to be deleterious to protein function. Therefore, we next assessed whether substitutions at detected vs undetected cysteines or lysines were more likely to be predicted damaging. We first assessed the amino acid substitutions for cysteine and lysine resulting in the greatest chemical property change, or highest Grantham score (Grantham, 1974), Cys > Trp and Lys > Ile. For CADD38 (Kircher et al, 2014), fathmm-MKL coding (Shihab et al, 2014), and DANN (Quang, Chen, & Xie, 2015), substitutions of detected cysteines were less likely to be predicted damaging compared to substitutions of undetected cysteines (Fig 3E, red; Dataset EV16). In contrast, substitutions of detected lysines were more likely to be predicted damaging compared to substitutions of undetected lysines (Fig 3E, blue; Dataset EV16). This trend for cysteine and lysine predicted deleterious score enrichment extended to all missense types (Fig EV3A; Dataset EV16).

We next tested if these trends would extend to clinically validated "pathogenic" and "likely pathogenic" missense mutations, as identified by the ClinVar database (Landrum et al, 2018). ClinVar is the gold standard repository of genomic variants associated with monogenic disorders. In total, the filtered ClinVar dataset contained 2,225 disease-associated missense variants that change from a cysteine (1,653 variants) or lysine (572 variants). We found no significant enrichment of disease-associated variants in detected over undetected cysteines (Fig 3F, red; Dataset EV17). In contrast, detected lysines showed a significant enrichment for disease-associated variants relative to undetected lysines (Fig 3F, light blue). Combining cysteine and lysine data revealed detected residues in general as more likely to harbor disease-associated mutations

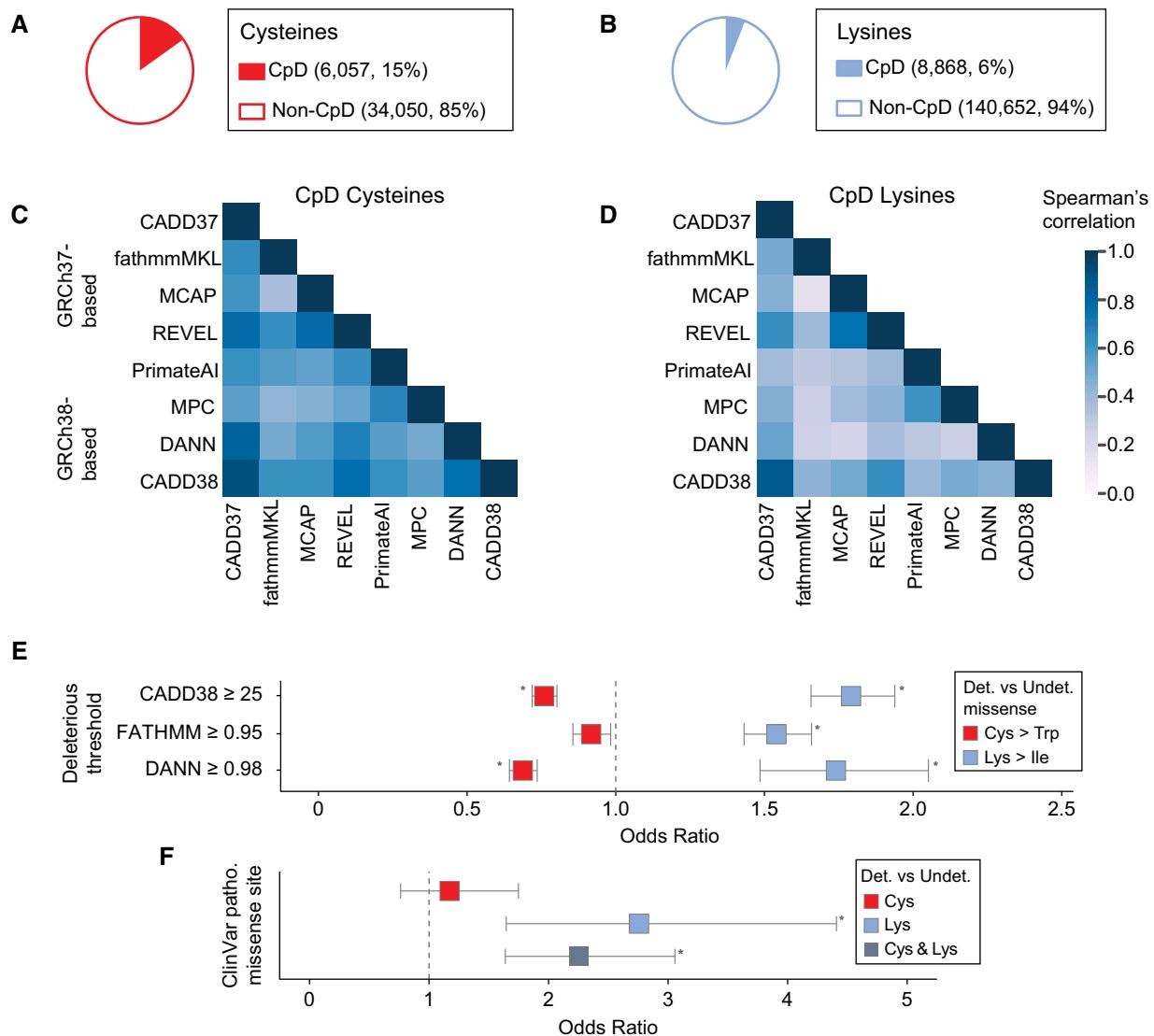

**Figure 3. Analysis of pathogenic missense at detected vs undetected cysteines and lysines.**

A, B   Aggregate number of detected and undetected cysteines (A) and lysines (B) in 3,840 CpDAA-containing proteins.

C   Heatmap of missense score correlations for all possible nonsynonymous SNVs at CpD cysteine (29,541 missense) for eight pathogenicity scores. Overall, Spearman's rank r was between 0.36 and 0.91.

D   CpD Lysine (41,850 missense) heatmap for missense score correlations for all possible nonsynonymous SNVs. Spearman's rank *r* between 0.16 and 0.81.

E   Odds of predicted deleterious Cys > Trp (red) missense at detected ($n$ = 6,057) vs undetected ($n$ = 34,049) residues in 3,840 detected proteins. Deleterious missense defined by CADD38, FATHMM, and DANN score thresholds ($y$ axis). CADD38 OR = 0.76, $P$ = 3.40e-22; FATHMM OR = 0.92, $P$ = 0.02; DANN OR = 0.690, $P$ = 6.69e-26. Odds of predicted deleterious Lys > Ile (blue) missense at detected ($n$ = 3,581) vs undetected ($n$ = 63,385) residues in 3,840 detected proteins. CADD38 OR = 1.80, $P$ = 1.03e-53; FATHMM OR = 1.55, $P$ = 3.47e-33; DANN OR = 1.75, $P$ = 9.21e-14. *$P$ < 0.0042 Bonferroni-adjusted (two-tailed Fisher's exact test).

F   Odds of ClinVar pathogenic variant overlapping detected (6,057 Cys; 8,868 Lys) vs undetected (34,050 Cys; 140,652 Lys) residues in 3,840 detected proteins. Cys detected in ClinVar pathogenic site (red, OR = 1.17, $P$ = 0.457) and Lys detected at ClinVar Pathogenic site (light blue, OR = 2.76, $P$ = 1.03e-04). Combined Cys and Lys (dark blue, OR = 2.26, $P$ = 9.99e-07) *$P$ < 0.0167 Bonferroni-adjusted (two-tailed Fisher's exact test).

Data information: In (E and F), 95% confidence intervals (line segments) and odds ratios (squares). In (C and D), color intensity represents two-tailed Spearman's rank-order correlation coefficients between 0 and 1.

relative to equivalent undetected residues in 3,840 detected proteins (Fig 3F, dark blue). Given the challenges associated with accurately diagnosing missense variants, we expect that chemoproteomic detection, particularly for lysine residues, could be used as an additional metric to improve pathogenicity predictions for genetic variants.

**Chemoproteomics data combined with pathogenicity scores can help prioritize functional residues**

We next assessed correlations between genetic-based pathogenicity score and amino acid reactivity, as assessed by chemoproteomics. We chose CADD as the optimal score to evaluate, as it integrates

other nucleotide variant predictors into its model and is available for both reference genome assemblies, GRCh37 and GRCh38. Chemoproteomic reactivity measurements were binned into low, medium, and high reactivity categories, defined as low ($R_{10:1} > 5$), medium ($2 < R_{10:1} < 5$), high ($R_{10:1} < 2$) isoTOP-ABPP ratios, respectively (Weerapana *et al*, 2010; Hacker *et al*, 2017). These ratios quantify the relative labeling of a residue at different probe concentrations (e.g., 1× vs 10×). A ratio closer to one indicates that labeling is saturated at low probe concentration, which corresponds to a cysteine or lysine with higher intrinsic reactivity.

To adapt CADD scores from the nucleotide level to the amino acid level for CpDAAs, the mean and max CADD score for all possible nonsynonymous SNVs per codon (see Methods) were calculated. For both max (Fig 4A) and mean (Fig EV3B) CADD codon scores, we found that highly reactive cysteines show significantly higher

predicted deleteriousness. In contrast, lysine reactivity did not correlate with predicted pathogenicity (Fig 4B and EV3C).

As the legacy cysteine reactivity dataset was relatively small (94 high reactivity cysteines in total), we next sought to verify these striking correlations, using a larger dataset. For this, we subjected lysates from the immortalized human T lymphocyte Jurkat cell line to isoTOP-ABPP reactivity profiling, comparing cysteine labeling with 10 or 100 μM iodoacetamide alkyne probe, as has been described previously (Weerapana *et al*, 2010). In aggregate, we identified 4,291 cysteines across five replicate experiments (~ 4-fold more cysteines than were assayed by (Weerapana *et al*, 2010)), including 322 high, 1,448 medium, and 2,247 low reactivity residues. A strong correlation (Pearson correlation coefficient = 0.5) was observed between values reported in our new dataset and those reported previously (Appendix Fig S9). This rich dataset (Dataset

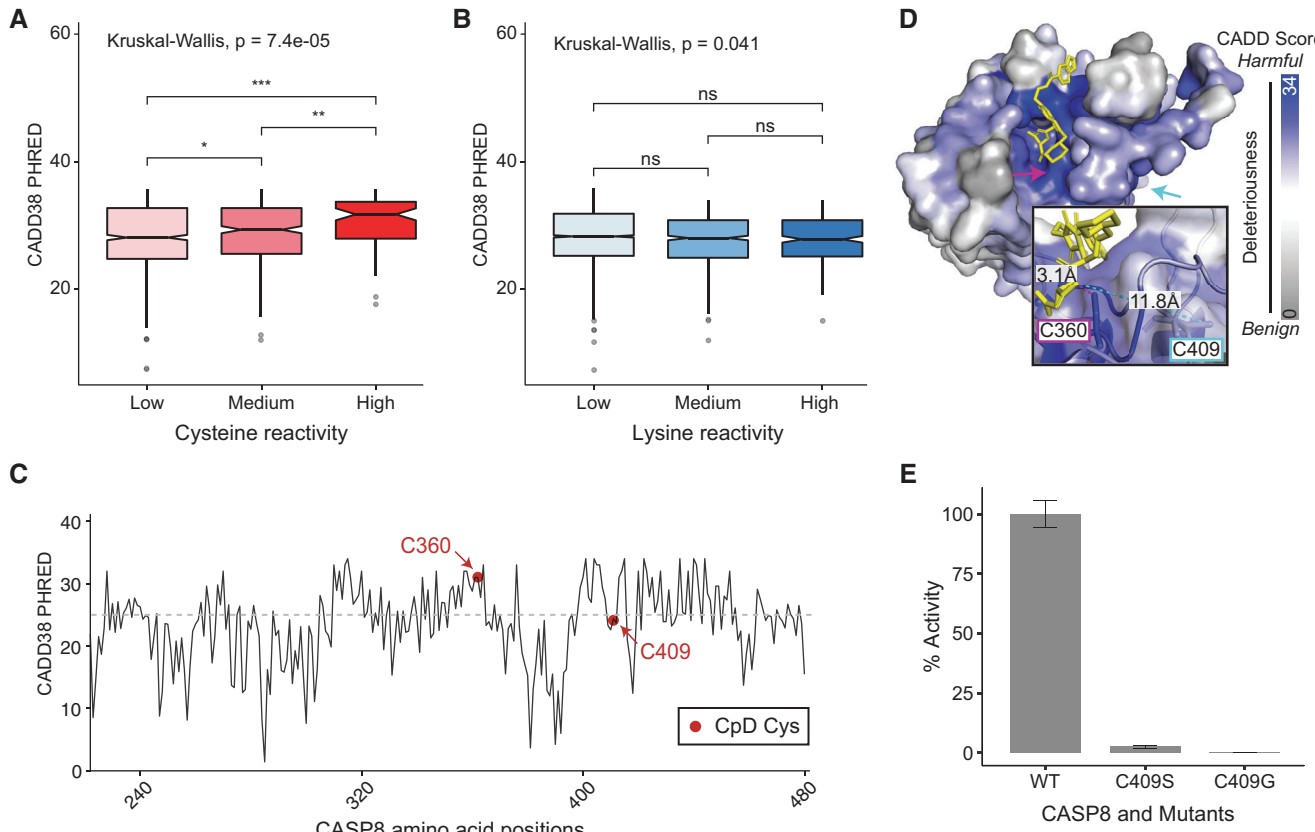

**Figure 4. Association between amino acid reactivity and CADD score.**

A, B   Distribution of the max CADD38 PHRED (model for GRCh38) scores for (A) cysteine (*n* = 1,401) and (B) lysine (*n* = 4,363) CpDAAs of low, medium, and high intrinsic reactivities, defined by isoTOP-ABPP ratios, low ($R_{10:1} > 5$), medium ($2 < R_{10:1} < 5$), high ($R < 2$) (Weerapana *et al*, 2010; Hacker *et al*, 2017). Kruskal–Wallis nonparametric test to examine reactivity group difference, and Wilcox test used for pairwise comparisons (BH-adjusted *P*-values, \**P*. adj = 0.04, \*\**P*. adj = 0.0037, and \*\*\**P*. adj = 0.00013). The boxplot boxes represent the lower and upper quartiles, with the central band as median. Notches show the confidence interval based on median ± 1.58\*IQR/sqrt(*n*), and whiskers mark observations that satisfy quartiles ± 1.5\*IQR. Median CADD38 max codon scores with bootstrapped 95% confidence intervals for reactive groups are low CpD Cys 27.3 (26.9, 28.0), medium CpD Cys 28.55 (27.80, 29.05), high CpD Cys 31 (28.8, 32.0), low CpD Lys 29.5 (29.3, 29.6), medium CpD Lys 29.25 (28.85, 29.50), high CpD Lys 29.05 (28.50, 29.55).

C   Shows CADD38 max codon scores for nonsynonymous SNVs at residues 220–479 of CASP8 (UniProt ID Q14790). Dashed horizontal line marks deleterious threshold of 25.

D   Crystal structure of CASP8 (PDB ID: 3KJN) highlighting C360 and C409. Bound covalent inhibitor B93 in yellow, with distance between Cys409 and the bound inhibitor measured in Angstroms. Protein surface color represents CADD38 max codon scores. Image generated in PyMOL (DeLano, 2002).

E   Activity of recombinant caspase-8 protein assayed using fluorogenic IETD-AFC substrate. Percentage activity shown relative to wild-type (WT) protein for three replicate experiments, bars and error bars as mean ± SD.

EV18) allowed us to further verify our finding that the codons of highly reactive CpDAAs are enriched for high pathogenicity scores. Gratifyingly, our initial finding was reproduced with this new and larger dataset (Appendix Fig EV3B and C), supporting both the validity of our approach and the robustness of our findings.

As a first case study to explore the utility of integrating genetic-based pathogenicity predictions with CpDAA reactivity measures, we turned to the well-characterized essential enzyme glucose-6-phosphate dehydrogenase (G6PD). Associated with over 160 different genetic mutations, G6PD deficiency is one of the most common genetic enzymopathies (Hwang *et al*, 2018). As G6PD deficiency is associated with both acute and chronic hemolytic anemia (Porter *et al*, 1964; Miwa & Fujii, 1996) (OMIM #300908), and with malaria resistance (Luzzatto, Usanga, & Reddy, 1969) (OMIM #611162), identifying functionally important residues in G6PD should inform the diagnosis and treatment of G6PD-associated genetic disorders. To visualize CADD pathogenicity scores along protein sequence length, we plotted the first 300 amino acids in G6PD with lines tracking max CADD GRCh38 scores, including the positions of all 15 residues identified in prior chemoproteomics studies (Fig EV4A). Of particular interest to us were K171 and K205, which are both located proximal to the enzyme active site (Fig EV4B). While K171 and K205 had very different intrinsic reactivities ($R_{10:1} = 1.3$ and $R_{10:1} = 9.2$, respectively), both showed high max CADD scores (28.8 and 32, respectively; Fig EV4A). Consistent with the observed high CADD scores, chemical modification at K205 (e.g., by aspirin) has been found to block G6PD activity (Jeffery, Hobbs, & Jörnvall, 1985; Ai *et al*, 2016) and mutations at K171 have been implicated in anemia (Hirono *et al*, 1989; Au *et al*, 2000). These prior data, when combined with our analysis of CADD and reactivity measurements support our finding that the propensity of lysines to react with electrophilic probes, but not measured differences in their intrinsic probe reactivity, correlate with predicted pathogenicity (Figs 3E and F, and EV3A and C).

We next sought to determine whether the utility of integrating genetic-based pathogenicity predictions with CpDAA reactivity measures could extend to the *de novo* discovery of functional residues. We turned to the well-characterized enzyme caspase-8, a member of the cysteine-aspartic acid protease (caspase) family and a key initiator of extrinsic apoptosis. Pathogenic mutations in caspase-8 result in autoimmune lymphoproliferative syndrome (ALPS, OMIM# 607271) (Chun *et al*, 2002; Kanderova *et al*, 2019) and are associated with certain types of cancer. Our chemoproteomic reactivity dataset (Dataset EV18 and Fig EV5A) revealed that caspase-8 harbors two iodoacetamide alkyne-reactive cysteines: the catalytic cysteine (Cys360, $R_{10:1} = 3.8$) and a second non-catalytic cysteine (Cys409, $R_{10:1} = 2.9$). Consistent with its function as the catalytic nucleophile, the codon of Cys360 has a high mean CADD score (29.3), whereas the codon of Cys409 has a lower CADD score (21.4), indicative that mutations that alter Cys409 should be less damaging to caspase-8 (Fig 4C). Cys409 is located on a flexible loop ~ 11.8 Å from the active site, as revealed by our projection of the max CADD codon scores onto the CASP8 X-ray structure (Fig 4D). As, to our knowledge, the functional impact of Cys409 mutations has not been assessed, we tested whether mutations at Cys409 would impact protein function, as indicated by the elevated measured reactivity, but not the moderate CADD score. Activity assays revealed that mutations at Cys409 do indeed impact protein

function, completely blocking proteolytic activity (Fig 4E; Dataset EV19). Taken together, these analyses highlight the utility of integration of chemoproteomic measures with pathogenicity predictions to improve stratification of functional and pathogenic residues.

## Discussion

We conducted an in-depth assessment of multiple mapping strategies to facilitate multi-omic analysis of chemoproteomics datasets. We then applied our optimal mapping strategy to analyze the relationship between missense pathogenicity scores and chemoproteomic measures of the intrinsic reactivity of cysteine and lysine residues. Our study revealed a number of challenges that limit the precision of multi-omic data analyses when using publicly available chemoproteomics datasets. To increase awareness of identifier mapping problems and to highlight important considerations for those analyzing similar datasets, we have summarized a list of best practices for accurate curation of functional annotations for CpDAA (Table 1).

The availability of raw proteomics data in public repositories (e.g., PRIDE (Côté *et al*, 2012), PeptideAtlas, and Panorama (Sharma *et al*, 2014)) might suggest an obvious solution to address the challenges associated with reprocessing published data: to re-search raw data using a new UniProtKB reference. However, reprocessing raw proteomic datasets can be both computationally expensive and time-limiting. An important alternative is to re-map the processed residue-level data to a release of UniProtKB that serves as the reference proteome for all functional annotations of interest, facilitating comparisons between annotated datasets. Complicating matters, providing the reference search databases (typically a custom UniProtKB FASTA file) alongside the raw proteomics files is not routine, and, although UniProtKB is updated monthly, only annual releases are maintained long-term in the database archives. Simply put, the original reference search sequences used in a chemoproteomics study may no longer be accessible for subsequent follow-up studies. Use of non-matched reference files can result in data loss and annotation errors, which may confound interpretation. For example, when we remapped legacy protein identifiers to multiple UniProtKB releases, we lost 28–199 of proteins, which ranges from 0.6 to 4.8% of the original total CpDAA proteins (Appendix Fig S1). While this may, at first glance, seem to be a paltry fraction of all data, these losses can still prove problematic when key proteins of interest (Dataset EV20) are lost due to database release differences.

There are several interconnected causes for our observed data loss at the protein level. The absence of protein isoform-specific identifiers in most proteomics search databases, particularly when combined with database updates to canonical sequences, can lead to mismapping, as shown for PRMT1 and FKBP7 (Fig 2A). The small number of UniProtKB sequences for which the canonical sequence is not the UniProtKB "-1" entry can also lead to further mismapping, especially when using mapping software that relies on this assumption (Fig 2C). Making reference FASTA files publicly available alongside raw data files is a relatively simple solution to facilitate data integration (Table 1A).

Reversing the central dogma to map protein identifiers back to transcript and gene identifiers and CpDAA positions back to

**Table 1.  Recommended best practices for inter- and intra-database integration for chemoproteomics datasets.**

| Entry | Recommendation |
|---|---|
| A | Support integration of quantitative chemoproteomics studies by (i) providing reference UniProtKB FASTA files alongside raw proteomic data files and (ii) including genomic coordinates for the codons of identified amino acids in the reference files |
| B | Perform proteomics database searches against reference database sequences that map to known transcript and gene coordinates (e.g., CCDS) |
| C | Perform sequence identity checks, which will identify and minimize mismapping caused by canonical sequence updates between UniProtKB releases |
| D | Map data to the appropriate genome assembly for downstream applications.<br>Genome assembly updates can introduce or refine genome resolution and in doing so alter the genomic coordinates of codons. Not all downstream pathogenicity predictors are compatible with both GRCh37 and GRCh38 (Appendix Table S14) |

transcript and genomic coordinates adds several additional layers of mapping complexity. Ensembl stable identifiers (gene, transcript, and protein), which are linked to UniProtKB stable identifiers are useful for facilitating this process. However, the number of redundant sequences maintained by Ensembl and the dynamic landscape of Ensembl entries across releases complicates the use of Ensembl stable IDs for inter-database mapping. For example, for the protein G6PD, across the five Ensembl releases investigated, we identified seven stable protein IDs, of which only one was consistently identical to the UniProtKB canonical sequence for G6PD (Fig 2D). The unsynchronized and frequent database update cycles are a cause of mismapping, which is particularly problematic for large-scale residue-level annotation projects (Fig 2E and Appendix Fig S1). Practically speaking, what this means is that a CpDAA from an available proteomics dataset could easily be mapped to the incorrect amino acid in an ENSP, followed by the incorrect transcript position, incorrect genomic coordinates, and incorrect pathogenicity score. Although there are a number of tools (e.g., TransVar and BISQUE (Zhou *et al*, 2015; Meyer, Geske, & Yu, 2016) that facilitate inter-database cross-referencing, their performance can be limited by all the challenges outlined above. An important and easily implementable solution to these problems is to search proteomics data against a highly curated reference file, such as the UniProtKB subset of cross-referenced CCDS proteins (Table 1B). Additionally, where possible, sequence identity checks should be performed to verify the mapping of identified residues (Table 1C).

Choice of reference genome further complicates data mapping. While many studies have now transitioned to GRCh38, many useful annotations, including variant-, sub-gene-, and gene-level metrics (e.g., MPC, PrimateAI, M-CAP, CCR, LOEUF), were built using GRCh37 genome assembly and are generally incompatible with the more recent GRCh38 genome assembly (Dataset EV13). As GRCh37 was frozen in 2014, mismapping can occur from invalid coordinates of proteomics datasets generated using newer reference proteomes based on GRCh38 coordinates. For many annotations, the solution to different genome assemblies is to "lift-over" annotations to the other genome assembly. However, not all functional annotations are compatible with liftover, as shown in Dataset EV13. Local sequence

alignment tools can be used to address problems when transitioning between GRCh37 and GRCh38 but can be challenging to scale genome-wide. The transition of all relevant annotations to the GRCh38 reference genome is ongoing and will address many of the aforementioned issues. However, this move is a substantial undertaking that requires rerunning of large-scale datasets and extensive quality control measures. To make full use of these scores, we recommend mapping proteomics data to genomic coordinates for both assemblies (Table 1D).

Together our analysis of inter-database mapping enabled us to compile a rigorously curated dataset of CpDAAs that mapped to both GRCh37- and GRCh38-based scores (data can be visualized in our CpDAA R Shiny app https://mfpalafox.shinyapps.io/CpDAA/). Using this dataset, we were then able to ask a number of novel questions, including how different scores compare across all identified cysteine and lysine residues and whether the codons of specific residues are enriched for predicted pathogenic mutations. For all nucleotide substitutions that result in a CpD cysteine or lysine amino acid change, we observed generally high concordance between scores (Fig 3C and D). While mutations at detected cysteine codons were, in general, predicted to be less deleterious than those at undetected cysteine codons, the subset of CpD cysteines with heightened reactivity were predicted to be more damaging than cysteines of lower reactivity (Figs 3E, 4A and EV3A). No such trend was observed for highly reactive lysines (Fig 4B; Appendix Fig S8). These intriguing findings suggest that cysteine hyper-reactivity is a privileged feature that could be used to inform the functions of genetic variants. As a demonstration of the utility of cysteine reactivity measures for identification of functional residues, we found that mutation of the non-catalytic Cys409 in caspase-8, which had an elevated reactivity ratio but relatively modest CADD score, completely ablated proteolytic activity, which supports that reactivity measurements likely can help to functionally stratify amino acids when CADD scores are less than conclusive.

We can foresee a multitude of applications for chemoproteomic and genomic data integration. While prior studies that revealed hyper-reactive cysteine residues are enriched in redox-active sites and enzyme active sites (Weerapana *et al*, 2010; Backus *et al*, 2016) and that hyper-reactive lysines were depleted in post-translational modification sites (Hacker *et al*, 2017), most CpDAAs still lack functional annotation. Predictive tools, such as those highlighted here, will undoubtedly aid in stratification of residues identified by chemoproteomics studies, pinpointing potentially druggable and disease-linked protein regions. To further aid in integration of CpDAA functional data, we have developed the CpDAA database to house all datasets used in this study together with their associated annotations.

Another area that we expect will benefit from such multi-omic approaches is interpretation of the impact of rare missense variants identified in patients with monogenic disorders. Protein-level functional data can aid in the interpretation of variants of uncertain significance (VUS), including those identified in clinical genetic testing, and can guide follow-up research studies. We anticipate that chemoproteomic methods should prove enabling for VUS interpretation, providing a high-throughput means to stratify amino acid functionality that is complementary to established genetic approaches, including site-directed mutagenesis. Application of chemoproteomics data to clinical studies will require careful data integration

and sequence level mapping, particularly given that the reference sequences and choice of identifiers employed by clinical vs research studies are typically non-identical.

Addition of protein structural data to such pipelines will likely further improve their utility and predictive power. As a starting point to such structure-based data integration, we mapped CADD predictive scores directly to the structures of CASP8 and G6PD (Fig 4D and EV4C). This 3-dimensional data integration highlighted key residues that form a common function in 3D space but are not easily identified using predictions associated with conservation in the linear-space of DNA. Looking to the future, we anticipate that such multi-omic studies will likely prove most enabling when combined with rigorous functional validation, for example by combining CRISPR-Cas9 mutagenesis with phenotypic assays. The use of CRISPR-Cas9 base editors (Kim *et al*, 2019; Grünewald *et al*, 2019, 2020) should facilitate such studies, particularly when combined with protein-centric guide RNA design packages (e.g., CRISPR-TAPE) (Anderson *et al*, 2020). In sum, we anticipate that such studies represent the next frontier for both the genetics and chemoproteomics communities.

# Materials and Methods

**Reagents and Tools table**

| Reagent/resource | Reference or source | Identifier or catalog number |
|---|---|---|
| **Experimental models** | | |
| Jurkat Clone E6, *Homo sapiens* | ATCC | TIB-152 |
| RPMI-1640 | Thermo Fisher | Cat# 11875119 |
| FBS | Thermo Fisher | Cat# 26400044 |
| **Oligonucleotides and other sequence-based reagents** | | |
| Primers for site-directed mutagenesis of Caspase 8 | | |
| C409G-F | CTGCTGGGGATGGCCACTGTGAATAACGGTGTTTCCTACCGAAACCCTGCAGAG | |
| C409G-R | CTCTGCAGGGTTTCGGTAGGAAACACCGTTATTCACAGTGGCCATCCCCAGCAG | |
| C409S-F | GTGAATAACTCTGTTTCCTACCGAAACCCTGCAGAGGGAAC | |
| C409S-R | GGAAACAGAGTTATTCACAGTGGCCATCCCCAGCAG | |
| **Chemicals, enzymes, and other reagents** | | |
| IAA | Backus *et al* (2016) | Supplementary Information (p.23) |
| TEV tags | Backus *et al* (2016) | Supplementary Information (p.24) |
| TEV protease | QB3 Macrolab, Berkeley | |
| Caspase-8 | Backus *et al* (2016) | Supplementary Information (p.11) |
| Caspase-8 activity assay | BioVision Incorporated | Cat# K112-100 |
| **Software** | | |
| Python Mapping Scripts | See GitHub for all scripts used in this paper https://github.com/mfpfox/MAPPING | |
| CpDAA Database | https://mfpalafox.shinyapps.io/CpDAA/ | |
| IP2 | http://www.integratedproteomics.com/ | |
| RAW Xtractor (version 1.1.0.22) | http://fields.scripps.edu/rawconv/ | |
| Prolucid | http://fields.scripps.edu/yates/wp/?page_id=17 | |
| DTASelect 2.0 | http://fields.scripps.edu/yates/wp/?page_id=816 | |
| Python 3.7.4 | https://www.python.org/ | |
| R 3.6.2 | https://www.r-project.org/ | |
| Tidyverse v1.3.0 | https://doi.org/10.21105/joss.01686 | |
| Pandas v0.25.1 | https://pandas.pydata.org/ | |
| Numpy v1.17.2 | https://numpy.org/ | |
| SciPy v1.3.1 | https://www.scipy.org | |
| Shiny | https://shiny.rstudio.com/ | |
| Prism 8 | GraphPad Software Inc. | |
| Adobe Illustrator | Adobe, Inc | |

**Reagents and Tools table**  (continued)

| Reagent/resource | Reference or source | Identifier or catalog number |
|---|---|---|
| **Data sources** | | |
| Ensembl releases | http://uswest.ensembl.org/info/data/ftp/index.html | |
| UniprotKB | https://www.uniprot.org/downloads | |
| dbSNFP v4.0a | https://sites.google.com/site/jpopgen/dbNSFP | |
| ClinVar database | https://www.ncbi.nlm.nih.gov/clinvar/ | |
| CADD v1.4 | https://cadd.gs.washington.edu/ | |
| fathmm-MKL | https://sites.google.com/site/jpopgen/dbNSFP | |
| DANN | https://cbcl.ics.uci.edu/public_data/DANN/ | |
| M-CAP v1.3 | http://bejerano.stanford.edu/mcap/ | |
| MPC release 1 | ftp://ftp.broadinstitute.org/pub/ExAC_release/release1/regional_missense_constraint/ | |
| REVEL | https://sites.google.com/site/revelgenomics/ | |
| Primate AI | https://sites.google.com/site/jpopgen/dbNSFP | |

## Methods and Protocols

### Data sources

All data sources are listed in Reagents and Tools table. CpDAA datasets were obtained from the following studies (Weerapana et al, 2010; Backus et al, 2016; Hacker et al, 2017). UniProtKB-SwissProt human proteome filtered by canonical isoform and cross-reference in CCDS database was downloaded August 06, 2018 (2018_06; see Reagents and Tools table). Two cross-reference file sources were used to map UniProtKB protein IDs to Ensembl IDs: (i) UniProtKB ID mapping (idmapping.dat) (McGarvey et al, 2019) or (ii) Ensembl release-specific mapping files (xref files) (Aken et al, 2016). ENSPs and identifiers were extracted from five release-specific FASTA files (Ensembl database version v85, v92, v94, v96, and v97) downloaded November 19, 2019. CADDv1.4 (Kircher et al, 2014) scores were downloaded on July 03, 2019. DANN (Quang, Chen, & Xie, 2015), fathmm-MKL (Shihab et al, 2014), M-CAP v1.3 (Jagadeesh et al, 2016), MPC release 1 (preprint: Samocha et al, 2017), REVEL (Ioannidis et al, 2016), and PrimateAI (Sundaram et al, 2018) scores were extracted from dbNSFPv4.0a (Liu et al, 2016) downloaded on June 11, 2019. "Pathogenic" and "likely pathogenic" labeled variants were extracted from the July 24, 2019, release of ClinVar (Landrum et al, 2018).

### Database update cycles

Average time between Ensembl, GENCODE, CCDS, and NCBI updates was quantified using all releases between August 2013 and July 2019 (5 years and 11 months window of time). Dates counted refer to the public release date posted on each databases' ftp site. For the UniProtKB update cycle length, values provided by the UniProtKB website on typical time between Knowledgebase releases from 2019 (4 weeks) and 2020 (8 weeks) were averaged. UniProtKB, Ensembl, GENCODE, CCDS, and NCBI releases were selected based on proximity to the release dates of the five Ensembl database versions analyzed in the current study.

### Mapping CpDAA data to more recent UniProtKB releases

CpDAA datasets had been previously searched against a non-redundant reverse concatenated UniProtKB reference FASTA file (Weerapana et al, 2010; Backus et al, 2016; Hacker et al, 2017) from the November 2012 (2012_11) release and amino acids in labeled peptides were annotated with the corresponding UniProtKB stable ID, amino acid letter, and position (e.g., P11413_C205). The author-provided UniProtKB 2012_11 FASTA file was referenced to check the UniProtKB IDs and CpDAA positions. Legacy chemoproteomic-detected cysteine and lysine positions that did not match positions in the canonical sequences from the 2012_11 release were dropped from further analysis. The UniProtKB 2012 canonical protein-based CpDAA residue numbers were then checked against UniProtKB canonical proteins from the 2018_06 release of CCDS cross-referenced human proteome dataset (See GitHub for python script). Chemoproteomic-detected proteins were excluded from further analysis if (i) UniProtKB canonical sequence from 2018 release was missing chemoproteomic-detected positions (e.g., natural variant overlaps detected cysteine position), (ii) UniProtKB ID flagged with "caution" on UniProt's website (e.g., https://www.uniprot.org/uniprot/Q8WUH1), and (iii) UniProtKB IDs not cross-referenced in all five Ensembl release-specific mapping files.

### Assessment of isoforms per stable UniProtKB ID

The UniProtKB homo sapien FASTA file containing canonical and isoform sequences was downloaded August 06, 2018. Isoform IDs per UniProt entry (referred to as stable ID in this study) were counted in the FASTA file. Canonical isoform IDs marked by lack of isoform name details (e.g., P11413) were excluded.

### Identification of UniProtKB canonical isoform ID numbers

UniProtKB canonical isoform ID numbers (e.g., P11413-X, "X" representing the isoform name) were identified for multi-isoform associated UniProtKB entries by comparing the 2018 UniProtKB FASTA file (used to count total isoforms per UniProtKB entry) and the UniProtKB ID mapping (idmapping.dat) file from August 01, 2018, release downloaded August 06, 2018. The FASTA file displays the canonical protein isoform ID with no isoform name details, but the idmapping.dat file displays the canonical isoform protein ID with these details.

### Inter-database identifier mapping (ID mapping) of CpDAA residues between UniProtKB and ENSPs

Two methods were used to cross-reference stable or versioned protein IDs between UniProtKB and five Ensembl releases:

### Method A

Ensembl mapping: Ensembl mapping ("xref") files from the five releases studied (v85, v92, v94, v96, and v97) were used for inter-database identifier mapping. Ensembl gene (ENSG), transcript, and associated protein IDs cross-referencing the curated set of 3,953 CpD UniProtKB stable IDs were extracted and grouped by single or multi-isoform status of the cross-referenced UniProtKB entry. Ensembl IDs cross-referencing UniProt CpD protein IDs were then used to filter the five Ensembl release-specific peptide FASTA files for associated protein sequences.

### Method B

UniProtKB isoform-specific mapping: UniProtKB ID mapping (idmapping.dat) file from August 01, 2018, release was used for inter-database identifier mapping. Ensembl IDs cross-referenced by the UniProtKB canonical protein isoform IDs for multi-isoform entries and stable IDs for single isoform entries were pooled and used to filter release-specific Ensembl peptide FASTA files for associated protein sequences.

### Assessing identifier multi-mapping between UniProtKB and Ensembl

From Method A ID mapping, the total number of unique Ensembl IDs (versioned and stable) from five releases that cross-reference CpD UniProt proteins was calculated for each UniProtKB ID. The mean number of unique multi-mapping Ensembl IDs per CpD UniProtKB protein ID was calculated for single and multi-isoform entries. Sequence identity was checked for all cross-referenced Ensembl and UniProtKB proteins and marked by an additional Boolean column ("False" for non-identical and "True" for identical Ensembl-UniProt canonical proteins; see GitHub for python script). From Method B ID mapping, as with analysis for Method A, identifier multi-mapping was calculated for single and multi-isoform UniProtKB entries and sequence identity of cross-reference proteins was marked by an additional Boolean column. Student's unpaired *t*-test was used to assess all ID multi-mapping differences between versioned and stable ENSG, transcript, and protein IDs cross-referencing our curated set of 3,953 CpD UniProt protein IDs found in all Ensembl release-specific mapping files.

### Identification of frequently updated Ensembl sequence types and non-identical cross-referenced UniProtKB-ENSPs

CpD UniProtKB canonical protein IDs were used to filter five Ensembl peptide FASTA files (Method A). A total of 8,861 unique Ensembl stable protein IDs were shared across all five Ensembl releases, cross-referencing a total of 3,887 CpD UniProtKB canonical proteins IDs. The 8,861 ENSP IDs with their associated stable gene and transcript IDs in each Ensembl release file were combined into a stable key ID (formatted as "ENSG_ENST_ENSP", for gene, transcript, and protein Ensembl stable IDs). Ensembl versioned IDs were additionally extracted from the release-specific FASTA files. To identify differences between ENSG, transcript, and protein sequence re-annotation rates, ID version number increments (signifying sequence re-annotation updates) relative to the v85 versioned IDs were summed for each ID biotype (gene, transcript, and protein ID extension numbers ".X"). To identify "dated" ID mappings, in which the cross-referenced ENSPs are no longer identical to canonical proteins from the 2018 UniProtKB release (current study's

reference proteome for CpDAA positions and functional annotations), sequence distance (IDs from Method B) was scored using the Hamming normalized distance metric (Frederick, Sedlmeyer, & White, 1993) and the Levenshtein normalized distance metric (Yujian & Bo, 2007). Normalized scale is 0 to 1, with 0 indicating identical Ensembl-UniProtKB proteins and 1 indicating significant differences between the two sequences.

### Residue mapping to pathogenicity scores

CpDAA-containing UniProt protein IDs and residue positions were mapped to dbNSFPv4.0a for annotations of missense deleteriousness scores. Additionally, undetected cysteine and lysine positions in CpD proteins were also pulled from dbNSFPv4.0. Genomic coordinate keys (formatted as "chr_pos_ref_alt") were made from the dbNSFP columns for GRCh37 and GRCh38 genome assemblies. Coordinate keys from dbNSFP were then used to map CADDv1.4 model annotation files. Missense overlapping cysteine and lysine codons in CpD proteins were required to have valid coordinates in both genome assemblies and annotations for all possible nonsynonymous SNVs (stop-gained missense consequences were filtered out from our analysis). The deleteriousness scores with no missing annotations for loss-of-cysteine and loss-of-lysine missense (CADD, fathmm-MKL, and DANN) were summarized by taking the max or mean of all nonsynonymous variants per cysteine and lysine codon in successfully annotated CpD proteins (see GitHub for python scripts).

### Correlation of deleteriousness scores for CpD cysteine and CpD lysine missense variants

Relationship between missense deleteriousness prediction scores and chemoproteomic detection was assessed by Spearman's rank-order correlation using the SciPy stats module both for CpDAAs and for non-detected cysteine and lysine residues. A nonparametric correlation test was chosen based on non-normal distributions of missense scores for cysteines and lysines in CpD proteins. All correlations are based on a subset of cysteine and lysine missense variants with no missing score annotations. CADD raw scores were used instead of the PHRED scores, with "CADD37" denoting raw score from the CADD GRCh37 model and "CADD38" denoting raw score from the CADD GRCh38 model.

### Enrichment analysis of predicted and known pathogenic missense variants for cysteine and lysine residues in detected proteins

For the analysis with predicted deleteriousness scores, cysteine and lysine residues from 3,840 successfully annotated CpD proteins were filtered for Cys > Trp and Lys > Ile specific substitutions. Deleterious missense thresholds were set as follows: CADD PHRED scores from the GRCh38 model (CADD38) greater than or equal to 25, fathmm-MKL scores greater than or equal to 0.95, and DANN scores greater than or equal to 0.98. For each group, an odds ratio (OR) along with the 95% confidence interval (CI) was calculated using Fisher's exact test on a 2 × 2 contingency matrix. Evidence for statistical significance of association was determined using the Bonferroni-adjusted *P*-value cut-off of 0.004. For the analysis with ClinVar "pathogenic" and "likely pathogenic" variants, the downloaded ClinVar variant data were filtered for loss-of-cysteine and loss-of-lysine missense consequences ($n = 2,225$ pathogenic variants) by parsing the Human Genome Variation Society Sequence

Variant Nomenclature column (HGVS, e.g., p.Cys36Trp). In total, 389 pathogenic variants overlapped the genomic coordinates of cysteines and lysines in 3,840 CpD proteins. For each group, an estimate of fold enrichment or odds ratio (OR), along with the 95% confidence interval (CI) was obtained using Fisher's exact test on a $2 \times 2$ contingency matrix. Evidence for statistical significance of association was determined based on the Bonferroni-adjusted *P*-value cut-off of 0.0167.

### Bootstrap analysis of CADD38 PHRED max codon scores

The bootstrapping procedure for calculating the 95% confidence interval of median CADD38 PHRED max codon scores and further characterizing the differences between low, medium, and highly reactive residues was performed as follows: original CADD38 max scores for each sub-group were resampled 20,000 times with replacement, with the median of each bootstrapped sample calculated. This process produced 20,000 samples with 895 low, 412 medium, and 94 high observations for CpD Cys, and 3,401 low, 660 medium, and 302 high observations for CpD Lys.

### Mapping deleteriousness scores to protein structures

For the UniProtKB canonical proteins G6PD (P11413) and CASP8 (Q14790), CADD GRCh38 model PHRED scores for missense overlapping all amino acid positions were extracted and summarized by taking the max or mean of all missense scores per residue. Scores for all residue positions were extracted from the dbNSFPv4.0a file (see Reagents and Tools table). After checking the canonical protein positions against the cross-referenced Protein Data Bank (PDB) ID pulled from the UniProtKB website (PDB ID 3KJN for CASP8 and 2BH9 for G6PD), residue max CADD PHRED scores were mapped to protein structure through assignment of scores as beta factor values of protein structure alpha carbons (GitHub for python script; Dataset EV21).

### IsoTOP-ABPP sample preparation and analysis

IsoTop-ABPP samples were prepared as described previously (Weerapana *et al*, 2010; Backus *et al*, 2016). Briefly, cells were harvested and lysed by sonication in PBS. Proteomes were adjusted to 1 mg/ml. Samples were labeled for 1 h at ambient temperature with either 10 or 100 μM iodoacetamide alkyne (IA-alkyne, 5 μl of 1 or 10 mM stock in DMSO). Samples were conjugated by CuAAC to either the light (fragment treated) or heavy (DMSO treated) TEV tags (10 μl of 5 mM stocks in DMSO, final concentration = 100 μM), with TCEP (10 μl of fresh 50 mM stock in water, final concentration = 1 mM), TBTA (30 μl of 1.7 mM stock in DMSO/*t*-butanol 1:4, final concentration = 100 μM), and $CuSO_4$ (10 μl of 50 mM stock in water, final concentration = 1 mM). After 1h, the samples were pelleted and the pellets sonicated in ice-cold methanol (500 μl) and combined pairwise. The pellets were solubilized in PBS containing 1.2% SDS (1 ml) with sonication and heating (5 min, 95°C) and any insoluble material was removed by an additional centrifugation step at ambient temperature (14,000 *g*, 1 min). Samples were then enriched on streptavidin resin (100 μl slurry) in PBS (10 ml) with rotating for 90 min. Beads were then washed (2× PBS and 2× water), resuspended in 6 M urea reduced (20 mM DTT), and alkylated (40 mM iodoacetamide). Samples were then diluted to 2 M urea and 6 μl (2 μg) reconstituted MS grade trypsin (Promega V5111) was added and the samples were allowed to digest overnight. The beads were

then pelleted, washed (3× PBS and 3× water), and then resuspended in 75 μl TEV buffer (50 mM Tris, pH 8, 0.5 mM EDTA, 1 mM DTT). 5 μl TEV protease (80 μM) was added and the reactions were rotated for 7 h at 29°C. The samples were then cleaned using Micro Bio-Spin columns, desalted using Pierce C18 100 μl bed zip-tips, concentrated by speedvac and reconstituted in 20 μl 5% ACN and 1% formic acid.

### Liquid chromatography tandem mass spectrometry (LC-MS/MS) analysis

The samples were analyzed by liquid chromatography tandem mass spectrometry using a Q Exactive™ mass spectrometer (Thermo Scientific) coupled to an Easy-nLC™ 1000 pump. Peptides were resolved on a C18 reversed phase column (3 μM, 100 Å pores), packed in-house, with 100 μm internal diameter and 18 cm of packed resin. The peptides were eluted using a 140-min gradient of buffer B in buffer A (buffer A: water with 3% DMSO and 0.1% FA; buffer B: acetonitrile with 3% DMSO and 0.1% FA) and a flow rate of 220 nl/min with electrospray ionization of 2.2 kV. The regular gradient includes 0–5 min from 1 to 5%, 15–130 min from 5 to 27%, 15–137 min from 27 to 35%, and 137–138 min from 35 to 80% buffer B in buffer A. Data were collected in data-dependent acquisition mode with dynamic exclusion (15 s), and charge exclusion (1, 7, 8, > 8) was enabled. Data acquisition consisted of cycles of one full MS scan (400–1,800 $m/z$ at a resolution of 70,000) followed by 12 MS2 scans of the nth most abundant ions at resolution of 17,500.

### Peptide and protein identification

The MS2 spectra data were extracted from a raw file using RAW Xtractor (version 1.1.0.22; available at http://fields.scripps.edu/rawconv/). MS2 spectra data were searched using the ProLuCID algorithm (publicly available at http://fields.scripps.edu/yates/wp/?page_id = 17 using a reverse concatenated, non-redundant variant of the Human UniProtKB database (release-2020_01). Cysteine residues were searched with a static modification for carboxyamidomethylation (+57.02146) and isoTOP differential modification at cysteine residues (+464.28595 for light and +470.29976 for heavy). Peptides were required to have at least one tryptic terminus, allowed one missed cleavage event and to contain the isoTOP modification. ProLuCID data were filtered through DTASelect (version 2.0) to achieve a peptide false-positive rate below 1%.

### Proteomic data processing

Custom python and R scripts were implemented to filter and compile labeled peptide datasets. Peptides with one tryptic terminus were filtered out before further analysis. Unique proteins, unique residues (cysteines or lysines), and unique peptide-spectrum matches (PSMs) were quantified for each dataset, using unique identifiers. Unique proteins were established based on UniProtKB protein ID. Unique residues were classified by an identifier consisting of a UniProtKB protein ID and the residue number of the modified cysteine/lysine; residue numbers were found by aligning the peptide sequence to the corresponding UniProtKB protein sequence. Unique peptides were found based on sequences containing modified residue location. If a peptide was labeled at multiple residues, an identifier was generated for each protein ID and modified residue location. IsoTOP-ABPP ratios from each experiment were averaged and reported with $\pm$ SD.

### Recombinant caspase-8 expression and purification

Recombinant caspase-8 (residues 217–479) without the CARD domain subcloned into pET23b (Novagen) with C-terminal $His_6$-affinity tags was expressed as has been described (Backus *et al*, 2016) previously. Site-directed mutagenesis (Liu & Naismith, 2008) was conducted as has been described previously, using the primers shown in the Reagents and Resources Table.

### Caspase-8 activity assay

Caspase-8 assay was conducted with CASP8 activity assay kit (BioVision; K112-100), following the manufacturer's instructions. Briefly, recombinant protein was diluted to 500 nM into assay buffer (50 µl/well in a 96-well plate) following which IETD-AFC substrate (4 mM stock in DMSO of IETD-AFC) was added to each well (5 µl stock diluted into 50 µl assay buffer for a final concentration of 200 µM substrate) and the samples were incubated at ambient temperature for 1 h. Caspase activity was measured from the increase in fluorescence (excitation 380 nm emission 460 nm). Experiments were performed in triplicate. Background was calculated from samples lacking the recombinant caspase.

## Data availability

Code for the mapping analysis and figures is available on the GitHub site https://github.com/mfpfox/MAPPING. All chemoproteomics datasets along with functional annotations are made available to download through the CpDAA database https://mfpalafox.shinyapps.io/CpDAA/, an R Shiny-based web interface. The mass spectrometry proteomics data have been deposited to the ProteomeXchange Consortium via the PRIDE partner repository with the dataset identifier PXD022151 (http://www.ebi.ac.uk/pride/archive/projects/PXD022151) and https://doi.org/10.6019/PXD022151.

**Expanded View** for this article is available online.

### Acknowledgements

This study was supported by a Beckman Young Investigator Award (Backus), V Scholar Award V2019-017 (Backus), and Chemistry Biology Interface Training Program T32GM008496 (Palafox) and DP5OD024579 (Arboleda). We thank Dennis W. Wolan and Gonzalo E. González-Páez for the recombinant caspase-8 proteins. We gratefully acknowledge all members of the Backus Lab and Arboleda Lab for their helpful suggestions.

### Author contributions

MFP, VAA, and KMB conceived and designed the study. MFP wrote code, analyzed data, and made figures. HSD generated chemoproteomics data. MFP, VAA, and KMB interpreted the results and wrote the manuscript.

### Conflict of interest

The authors declare that they have no conflict of interest.

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
