## [Review Process File · Molecular Systems Biology]

From Chemoproteomic-Detected Amino Acids to Genomic Coordinates: Insights into Precise Multi-omic Data Integration

Maria Palafox, Heta Desai, Valerie Arboleda, and Keriann Backus

DOI: [10.15252/msb.20209840](https://doi.org/10.15252/msb.20209840)

Corresponding author(s): Keriann Backus (kbackus@mednet.ucla.edu), Keriann Backus (kbackus@mednet.ucla.edu), Valerie Arboleda (VArboleda@mednet.ucla.edu)

Review Timeline:

Submission Date:	3rd Jul 20
Editorial Decision:	17th Aug 20
Revision Received:	20th Nov 20
Editorial Decision:	23rd Dec 20
Revision Received:	15th Jan 21
Accepted:	18th Jan 21

Editor: Maria Polychronidou

Transaction Report:

Thank you again for submitting your work to Molecular Systems Biology. We have now heard back from the three referees who agreed to evaluate your study. As you will see below, the reviewers acknowledge that you address a timely topic. However, they think that the study remains somewhat preliminary and raise a series of concerns, which we would ask you to address in a major revision.

Without repeating all the points listed below, some of the more fundamental issues are the following:

- Both reviewers #1 and #3 think that in its current form the study seems preliminary. During our pre-decision cross-commenting process (in which the reviewers are given the chance to make additional comments, including on each other's reports), reviewer #1 mentioned that they agree with reviewer #3 that the results and their significance are rather thin and that the second half of the manuscript has potential that needs to be more fully fleshed out. In line with this, and with the comments of reviewer #3 we would ask you to include validations of (some of) the reported predictions and to provide concrete evidence that the combination of the different data types can aid new discoveries.
- As reviewer #1 recommends, a clear message should be extracted from the analyses reported in figures 1-3, as well as actionable points and a potential workflow/code for future analyses.

All other issues raised by the referees would need to be satisfactorily addressed. Please let me know in case you would like to discuss in further detail any of the issues raised.

On a more editorial level, we would ask you to address the following issues.

REFeree REPORTS

Reviewer #1:

Reviewer comments for manuscript entitled "From Chemoproteomic-Detected Amino Acids to Genomic Coordinates: Insights into Precise Multi-omic Data Integration".

Summary:

In this manuscript the authors seek to actively address the widely experienced challenge of cross-platform multi-omic data integration. Specifically, problems arising during the mapping of legacy -omic scale data with new database releases. They focus on the integration and mapping of previously published mass-spectrometry datasets with versions of the human genome. The unstable nature of identifiers is discussed in detail, and provided in historical context for different database releases since 2012. The authors then produce a carefully curated dataset of previously published proteins containing chemoproteomic detected amino acids (CpDAAs) to correlate amino acid reactivity with predicted pathogenic variant data (CADD and others) available for the genomic datasets.

Overall the manuscript is well written, with high quality figures. While the data provided support their conclusions, the overall use of some portions of the analysis are unclear. The handling and integration of datasets in the era of "big data" is a well-recognized problem, and the authors spend considerable effort demonstrating the challenge of the integrating cross-platform multi-omics datasets. However, they do not seem to provide a clear solution. The discussion of the problem, and emphasis on the need for careful curation (as no other solution seems to be forthcoming) covers the first three multipanel figures. This leads to the second point that the argument that this is a challenge is overstated - is this really a five-figure manuscript? Is the level of insight provided into an appreciated problem worth three figures? As it is currently written it is not clear if the authors do in fact arrive at a solution to the mapping issue described. The methods include vague references to custom code, without descriptions or details of what the code enables them to achieve. Do these codes provide the solution to the problem with mapping legacy data to current database releases? If so, then this would be of great use to the field! It should be more fully described and their application made more accessible and user-friendly than basic github deposition of the code.

While the first section seems somewhat unnecessary, the latter half of the manuscript and datasets contained therein (figures 4 and 5) provide some interesting analysis and insight. Particularly in regard to the increasingly recognized challenge of target (and in this case amino acid) prioritization for further study and drug discovery.

On the basis of the above I recommend that the manuscript could be accepted following major revisions and re-review.

Major comments:

- If the authors do have a solution to problems arising from cross-platform multi-omic integration of legacy data, this should be more clearly emphasized in the text and with a figure, with the code/method/protocol/custom-processing algorithm more clearly described and provided as a resource for researchers encountering similar problems.
- If the authors have not developed a code-based or other reliable method/protocol solution beyond basic careful curation we would suggest that the manuscript be redrafted and reconfigured to a shorter format. Specifically, I would suggest that the entire first section (figures 1, 2 and 3) be moved to the supplemental/extended material. Should the authors still wish to present some of these data as a main figure the salient points should be condensed into a one single figure. Alternatively, can the data in figures 1-3 be better used to provide greater insight into the important question of whether or not key pathogenic variants have been missed in relation to the legacy proteomic data i.e. what are the real-world consequences and implications of the versioning issues identified?
- The authors could make better use of their curated dataset - they map their reactive sites to predicted pathogenicity scores. While this analysis is useful and provided in a useful format that supports visualization and interpretation, why not map them to known clinical variants available through publically available resources such as ClinVar? This would be exciting to see!
- Page 16: "Missense changes with high Grantham score (Grantham 1974), Cys>Trp and Lys>Glu, were used to assess the impact of these non-synonymous substitutions (missense) at detected versus undetected cysteine and lysine residues." The rationale provided for choosing these mutations is unclear and should be expanded. According to the Grantham scoring method referenced, the Cys>Trp score is the highest of all possible amino acid pairings with cysteine. However, Lys>Glu is not. According to the matrix provided in that reference Lys>Cys, Tyr, Phe, Ile, Gly, Val, Ala, Thr, Pro, Leu, Ser, Asn, Trp, Met, Asp all score more highly than the Lys>Glu mutation selected. According to the justification for the selection, this does not then make sense. Could the authors please explain this in more detail.

Minor comments:

- Page 6: "These 15,617 CpDAAs are further sub-categorized by the measures of amino acid intrinsic reactivity and potential targetability" - please provide details regarding the method used to determine targetability.
- Page 6-7: "Therefore, we evaluated established methods for inter-database mapping, including ID mapping, residue-residue mapping, and residue-codon mapping (See Table S2 for detailed descriptions of each type of mapping)." The descriptions in table S2 are not detailed, and do not provide references for these "established methods" and so the methods themselves cannot be fairly assessed. Please provide further details and references.
- Figure 4 legend: "Cys>Trp (yellow) and Lys>Glu (purple)" - this is not the color scheme used on the figure, please correct.

Reviewer #2:

In this well-written manuscript, Palafox et al. detail the challenges of accurate amino acid residue mapping, along with translation to transcript or gene sequence, in databases that are frequently updated. As they mention, while the number of proteins that can be lost in comparing different databases is small, the potential impact of these lost proteins can be large if one or more of the proteins is critical in the biology of the system. I found their analysis to be very thorough and I believe their findings will be highly interesting to the systems biology community. As one solution suggested by the authors, storage of the FASTA file along with the raw proteomics data would enable reproducible analysis of the proteomics data without concern over different versions of the reference data. After a thorough comparison of different databases and their effects on proteins with different characteristics, the authors turn to cysteine and lysine residues from chemoproteomic datasets, and evaluate their reactivity vs. pathogenicity scores. Application of this analysis to G6PD highlights a couple of lysine residues in the active site of the protein, suggesting two sites for future follow-up studies. Overall this manuscript is timely, of general interest to a broad audience, and highlights the potential for using pathogenicity scores to direct genomic mutations.

Comments:

1. No reference to Figure 4D in the text.
2. In figure 5C there are two other lysine residues that have CADD scores >25, yet neither are highlighted nor discussed. What are these residues, where do they fall on the structure, and how does this affect interpretation of the approach?
3. In Figure 5D, Lys171 may be directly interacting with NADP+, but lys205 appears to be outside of the direct interaction space. A different visualization may be helpful to provide a more atomic / molecular perspective on the role of these residues.
4. In Figure 5A and 5B, was multiple hypothesis testing used in assessing significance?

Reviewer #3:

The manuscript integrates chemoproteomic data with predictions of genetic variant pathogenicity, to identify codons of highly reactive cysteines that are enriched for genetic variants that are predicted to be more deleterious. According to the authors this will advance prioritization of putative druggable sites. Specifically This manuscript describes variability between data found in Ensembl, GENCODE, NCBI, and CCDS. They propose using version IDs or a cross reference file from UniProtKB that displays the CCDS canonical isoform ID mapped to a stable Ensembl protein ID to map chemoproteomic data between databases. They then mapped the CpDAA coordinates to a panel of functional annotation scores and calculated the correlation of scores for overlapping codon coordinates of CpD cysteine and lysine residues. While this paper has some interesting preliminary results I don't think it is of sufficient significance to be published in a journal like MSB.

1. The most significant issue is that the major conclusion is not supported by the data. The authors write "Taken together, this analysis highlights the utility of integration of pathogenicity predictions to improve stratification of chemoproteomics data." (page 18). This is a strong claim especially since there is no experimental validation for any of the predictions. It would be critical to test some of the predictions experimentally.

2. Another important issue is that the results and their significance are very thin. While the generation of the database by combining different data types should be commended, the analysis of this set is very limited. The results section focuses too much on how the data was integrated

and not on what we can learn from it.

3. There are also significant problems with the writing of the manuscript. It is hard to distinguish between the results and methods. For example, Pages 5-8 read more like Methods and make the manuscript hard to read. The results in section 3 (pages 11-14), are also very descriptive and read like methods.

4. There are also serious organizational issues with the manuscript. For example, in page 7, second paragraph "Specific releases were prioritized... ". The manuscript does not describe the importance or reason for this criteria when it is first introduced. It does include some information about (1) on page 15. This delayed partial explanation makes the selection criteria unclear for a large portion of the manuscript.

5. Page 8. "We assembled a comprehensive inventory of all database updates for Ensembl, GENCODE, NCBI, and CCDS (Table S4) between the dates of August 2013 - July 2019 (6 years)". This paragraph does not describe the quantitative results from this analysis, but instead states that changes accumulate causing data loss.

6. Page 11. "ID cross-reference files (Table S2) that are released by Ensembl and UniProtKB ". What is the rationale for selecting G6PD for investigation? While there is a small description of clinical manifestations of G6PD deficiency on page 17, much like the comments about page 7 the placement seems delayed.

7. Page 12. The manuscript recommend using version IDs and then say why they are too challenging to use in the same paragraph

8. Page 13. The following statement is confusing. "To assess how pervasive multi-mapping is across the whole CpDAA dataset, we quantified the mean number of Ensembl IDs per UniProtKB IDs. We found that the mean number of Ensembl IDs per UniProtKB IDs was significantly increased in versioned IDs compared to stable IDs, both for Ensembl gene and transcript IDs ". Isn't that redundant- by having version IDs you increase the number of IDs?

9. Page 13 last paragraph. "We identified the top stable IDs ...". This analysis identified 49 UniProtKB IDs, is this a significant number?

10. Page 15 second paragraph. "When comparing only machine learning-based scores for substitutions ..." Only place the term machine learning is used in the manuscript. What do you mean by machine learning here? It doesn't feel like it fits in the document.

Point-by-Point Response to Summary Points:

1. Both reviewers #1 and #3 think that in its current form the study seems preliminary. During our pre- decision cross-commenting process (in which the reviewers are given the chance to make additional comments, including on each other's reports), reviewer #1 mentioned that they agree with reviewer #3 that the results and their significance are rather thin and that the second half of the manuscript has potential that needs to be more fully fleshed out. In line with this, and with the comments of reviewer #3 we would ask you to include validations of (some of) the reported predictions and to provide concrete evidence that the combination of the different data types can aid new discoveries.

We thank the reviewers for these constructive suggestions. To address these points and to further enhance the second half of the manuscript we have added the following key analyses, which we believe substantially strengthen our manuscript: First, we added additional analysis of CpDAAs identified in Clinvar, which revealed that chemoproteomic detected lysines showed significant enrichment of disease-associated variants (**Fig 3F**, blue). Second, we generated additional new chemoproteomic cysteine reactivity datasets, which identified in aggregate 4017 cysteines, of which 322 were found to be hyperreactive (**Fig EV5**). When compared to the prior published dataset (Weerapana et al., Nature 2010), which we initially analyzed, our new dataset represents a significant technical advance, as it represents a four-fold increase in the number of cysteines quantified, when compared with the prior study. This rich dataset allowed us to further verify that our exciting finding that the codons of highly reactive CpDAAs are enriched for high pathogenicity scores. Gratifyingly, our initial finding was reproduced with this new and much larger dataset (**Fig EV5A**), supporting both the validity of our approach and the robustness of our findings. Third, we conducted functional studies on the cysteine protease caspase-8, which provide additional concrete evidence that the combination of CpDAA data with predictions of pathogenicity can aid new discoveries. Our chemoproteomic reactivity dataset (**Dataset EV18** and **Fig EV5C**) revealed that caspase-8 harbors two reactive cysteines (the catalytic cysteine Cys360) and a second non-catalytic cysteine (Cys409). Consistent with its function as the catalytic nucleophile, the codon of Cys360 has a high mean CADD score (29.3), whereas the codon of Cys409 has a lower CADD score (21.4), indicative that mutations that alter Cys409 should be less damaging to caspase-8. We therefore sought to test whether mutations at Cys409 would impact protein function, as indicated by the elevated measured reactivity, but not the moderate CADD score. Functional assays revealed that mutations at Cys409 do indeed impact protein function, completely blocking proteolytic activity (**Fig 4E**). These data support that the reactivity measures, which indicated that this residue is likely functional, can complement the CADD data to more accurately pinpoint functional amino acids.

2. As reviewer #1 recommends, a clear message should be extracted from the analyses reported in figures 1-3, as well as actionable points and a potential workflow/code for future analyses.

We have consolidated Figures 1-3, which we believe improves the clarity and accessibility of the central message. We have also consolidated the corresponding results section for these figures for clarity. We have further added an additional table (**Table 1**), which summarizes the key take home message of the mapping portion of the paper and provides actionable strategies for improved inter-datatype mapping. We have also included an additional README document available at <https://github.com/Mfpfox/MAPPING> to help users work with their data sets to accurately map genomic features. Lastly, in addition to providing all code on GitHub, the "CpDAA Database" is now housed as an R Shiny App (<https://mfpalafax.shinyapps.io/CpDAA/>),

which allows readers to further visualize and explore the exciting correlations between chemoproteomic and pathogenicity prediction datasets.

Editorial Points:

1. Please include 5 keywords.

The following five keywords have been added: Chemoproteomics • Multi-omics • Genetic Pathogenicity Prediction • Amino Acid Reactivity • Inter-database Mapping

2. Please provide a .doc version of the manuscript text (including legends for main figures and tables) and individual production-quality files for the main figures.

A .doc version of the manuscript has been submitted, including all figures and tables, together with 300 dpi versions of all main figures.

3. We have replaced Supplementary Information by the Expanded View (EV format). In this case, all additional figures can be included in a PDF called Appendix. **Appendix figures and Tables should be labeled and called out as: "Appendix Figure S1, Appendix Figure S2... Appendix Table S1..." etc.** Each legend should be below the corresponding Figure/Table in the Appendix. Please include a **Table of Contents** in the beginning of the Appendix. For detailed instructions regarding expanded view please refer to our Author

We have followed the Author guidelines and have included both EV and Appendix Figures. A Table of Contents has been included in the Appendix.

4. Tables S1-S16 need to be provided as Datasets EV1-EV16 (**one file per dataset**). Please include in each of the .xlsx files, a **description of the dataset in a separate tab**.

Tables S1-S16 are now provided as Datasets EV1-EV21, which include descriptions of the dataset in a separate tab.

5. Please provide a "standfirst text" summarizing the study in one or two sentences (approximately 250 characters), **three to four "bullet points" highlighting the main findings and a "synopsis image"** (550px width and max 400px height, jpeg format) to highlight the paper on our homepage.

The following standfirst text has been added:

Summary: We develop a multi-omic data integration workflow to map Chemoproteomic Detected (CpD) amino acids to genomic-level predictions of variant pathogenicity. Highly reactive cysteine and lysine residues are enriched for high pathogenicity (CADD) scores and pathogenic variants that cause clinical disease.

- Comparison of multi-omic mapping strategies identifies common issues with data integration, including those that result from updates to and redundancy of reference sequences.

- Chemoproteomic detected cysteines showed no significant enrichment of disease-associated variants in CLINVAR, whereas detected lysines showed significant enrichment for pathogenic variants.
- Proof-of-concept functional validation studies reveal that chemoproteomics measures of cysteine reactivity complement genetic scores to accurately predict functional residues in the cysteine protease caspase-8.

A synopsis image has been provided.

6. All **Materials and Methods** need to be described in the main text. We would encourage you to use 'Structured Methods', our new Materials and Methods format. According to this format, the Material and Methods section should include a Reagents and Tools Table (listing key reagents, experimental models, software and relevant equipment and including their sources and relevant identifiers) followed by a Methods and Protocols section in which we encourage the authors to describe their methods using a step-by-step protocol format with bullet points, to facilitate the adoption of the methodologies across labs.

All materials and methods have been included in the main text and a Reagents and Tools Table has been added.

7. Please use the **Data availability section** to describe how the data and code have been made available. This section needs to be formatted according to the example below:

The **Data and source code availability section** of the Material and Methods section now includes the following:

Data and code are available at the github site below are sufficient to reproduce the plots and analyses in this paper are available at <https://github.com/mfpfox/MAPPING>. An interactive version of the CpDAA datasets with all chemoproteomic data included in this study is available at our CpDAA database. <https://mfpalafox.shinyapps.io/CpDAA/>. The mass spectrometry proteomics data have been deposited to the ProteomeXchange Consortium via the PRIDE partner repository with the dataset identifier PXD022151 and 10.6019/PXD022151.

- **For data quantification:** please specify the name of the statistical test used to generate error bars and P values, the number (n) of independent experiments (specify technical or biological replicates) underlying each data point and the test used to calculate p-values in each figure legend. The figure legends should contain a basic description of n, P and the test applied. Graphs must include a description of the bars and the error bars (s.d., s.e.m.).

Figure legends now contain descriptions of n, p and the test applied. Graphs now include a description of the bars and error bars.

- When you resubmit your manuscript, please download our **CHECKLIST** (<http://bit.ly/EMBOPressAuthorChecklist>) and include the completed form in your submission.

Checklist has been included.

A Point-by-Point response to all reviewer comments:

Reviewer #1:

Reviewer comments for manuscript entitled "From Chemoproteomic-Detected Amino Acids to Genomic Coordinates: Insights into Precise Multi-omic Data Integration".

Summary:

In this manuscript the authors seek to actively address the widely experienced challenge of cross-platform multi-omic data integration. Specifically, problems arising during the mapping of legacy -omic scale data with new database releases. They focus on the integration and mapping of previously published mass-spectrometry datasets with versions of the human genome. The unstable nature of identifiers is discussed in detail, and provided in historical context for different database releases since 2012. The authors then produce a carefully curated dataset of previously published proteins containing chemoproteomic detected amino acids (CpDAAs) to correlate amino acid reactivity with predicted pathogenic variant data (CADD and others) available for the genomic datasets.

Overall the manuscript is well written, with high quality figures. While the data provided support their conclusions, the overall use of some portions of the analysis are unclear. The handling and integration of datasets in the era of "big data" is a well-recognized problem, and the authors spend considerable effort demonstrating the challenge of the integrating cross-platform multi-omics datasets. However, they do not seem to provide a clear solution. The discussion of the problem, and emphasis on the need for careful curation (as no other solution seems to be forthcoming) covers the first three multipanel figures. This leads to the second point that the argument that this is a challenge is overstated - is this really a five-figure manuscript? Is the level of insight provided into an appreciated problem worth three figures? As it is currently written it is not clear if the authors do in fact arrive at a solution to the mapping issue described. The methods include vague references to custom code, without descriptions or details of what the code enables them to achieve. Do these codes provide the solution to the problem with mapping legacy data to current database releases? If so, then this would be of great use to the field! It should be more fully described and their application made more accessible and user-friendly than basic github deposition of the code.

While the first section seems somewhat unnecessary, the latter half of the manuscript and datasets contained therein (figures 4 and 5) provide some interesting analysis and insight. Particularly in regard to the increasingly recognized challenge of target (and in this case amino acid) prioritization for further study and drug discovery.

On the basis of the above I recommend that the manuscript could be accepted following major revisions and re-review.

We are delighted that this reviewer found our manuscript well written, and we thank them for their careful analysis of our manuscript. To address the reviewer's concern about the absence of a clear solution for the inter-datatype mapping, we have added an additional table (**Table 1**), which summarizes the key actionable strategies for improved inter-datatype mapping. In addition to providing all source code on GitHub, we have developed an database that integrates the CpDAA data from this paper into a searchable browser already integrated with functional annotation. This new app, "CpDAA Database," is housed as an R Shiny app and currently hosts 16,999 CpDAA mapped to the gene and residue-level functional annotations and is currently hosted at <https://mfpalafax.shinyapps.io/CpDAA/>. This App allows readers to further visualize and explore the exciting correlations between chemoproteomic and pathogenicity prediction datasets.

To address the reviewer's concern that the first portion of the manuscript overstated the challenge of inter-datatype mapping, we have consolidated Figures 1-3, which we believe improves the clarity and accessibility of the central message. We have also consolidated the corresponding results section for these figures for clarity.

To address this reviewer's concern about the useability of our code we have also included an additional README document available at <https://github.com/Mfpfox/MAPPING> to help users work with their data sets to accurately map genomic features.

Major comments:

- *If the authors do have a solution to problems arising from cross-platform multi-omic integration of legacy data, this should be more clearly emphasized in the text and with a figure, with the code/method/protocol/custom-processing algorithm more clearly described and provided as a resource for researchers encountering similar problems.*

We have added **Table 1**, which summarizes the key actionable strategies for improved inter-datatype mapping. We have also added an additional protocol (provided as a README with source code) to serve as a resource for researchers who encounter similar problems.

- *If the authors have not developed a code-based or other reliable method/protocol solution beyond basic careful curation we would suggest that the manuscript be redrafted and reconfigured to a shorter format. these data as a main figure the salient points should be condensed into a one single figure. Alternatively, can the data in figures 1-3 be better used to provide greater insight into the important question of whether or not key pathogenic variants have been missed in relation to the legacy proteomic data i.e. what are the real-world consequences and implications of the versioning issues identified?*

While we have added additional code and a protocol, we agree with this reviewer that the first half of the manuscript was overly long and confusing. To address this limitation, we have condensed Figure 1 and consolidated Figures 2 and 3. The corresponding text has also been significantly shortened, and we have moved several methods-related sections to the methods section. We believe that these changes make the significance of our findings and overall message more accessible to the reader. We also appreciate the reviewer's suggestion of including known pathogenic missense variants in our analysis.

To this end, the new Figure 3 has been updated to include the Fisher's Exact odds ratio for pathogenic and likely pathogenic ClinVar mutations at detected versus undetected cysteine and lysine residues (**Fig 3F**). We also have included on the R Shiny app homepage a table with all ClinVar pathogenic and likely pathogenic variants that overlap the codon coordinates of CpDAA residues. Surprisingly, we found that while detected lysines were enriched for known pathogenic variants, detected cysteines were not. These findings hint that buried cysteines and those found in structural disulfides may be enriched for pathogenic mutations, which are exciting models that we expect will form the foundation for future studies in this space.

- *Page 16: "Missense changes with high Grantham score (Grantham 1974), Cys>Trp and Lys>Glu, were used to assess the impact of these non-synonymous substitutions (missense) at detected versus undetected cysteine and lysine residues." The rationale provided for choosing these mutations is unclear and should be expanded. According to the Grantham scoring method referenced, the Cys>Trp score is the highest of all possible amino acid pairings with cysteine. However, Lys>Glu is not. According to the matrix provided in that reference Lys>Cys, Tyr, Phe,*

Ile, Gly, Val, Ala, Thr, Pro, Leu, Ser, Asn, Trp, Met, Asp all score more highly than the Lys>Glu mutation selected. According to the justification for the selection, this does not then make sense. Could the authors please explain this in more detail.

We thank the reviewer for noting this discrepancy. To address this point, we have updated the figure to show Lys>Ile instead of the originally selected Lys>Glu, as shown in the new **Fig 3E**. When considering only substitutions possible by single nucleotide variants (SNVs), isoleucine is the highest Grantham distance compared to lysine. It is worth noting that, as isoleucine is only encoded by one codon, this analysis is less powered than our previous analysis of Cys>Trp substitutions. We have therefore also included in **Fig EV3** the Fisher's Exact odds ratios for all cysteine and lysine substitutions possible from nonsynonymous variants, which reveals similar trends across all possible substitutions at cysteine and lysine codons.

Minor comments:

- *Page 6: "These 15,617 CpDAAs are further sub-categorized by the measures of amino acid intrinsic reactivity and potential targetability" - please provide details regarding the method used to determine targetability.*

We agree with this reviewer that our original phrasing was unnecessarily confusing. To address this point, we have modified the text to say "These 15,617 CpDAAs are further sub-categorized by the residues labeled by cysteine- or lysine-reactive probes (iodoacetamide alkyne [IAA] or Pentynoic acid sulfo tetrafluorophenyl ester [STP], respectively) and those residues with additional measures of intrinsic reactivity (categorized of high-, medium-, and low-reactive residues)."

- *Page 6-7: "Therefore, we evaluated established methods for inter-database mapping, including ID mapping, residue-residue mapping, and residue-codon mapping (See Table S2 for detailed descriptions of each type of mapping)." The descriptions in table S2 are not detailed, and do not provide references for these "established methods" and so the methods themselves cannot be fairly assessed. Please provide further details and references.*

Further details and references have been added to Table S2.

- *Figure 4 legend: "Cys>Trp (yellow) and Lys>Glu (purple)" - this is not the color scheme used on the figure, please correct.*

We thank the reviewer for noting this discrepancy. This figure legend (now for the new Fig 3) has been changed to reflect the color scheme.

Reviewer #2:

In this well-written manuscript, Palafox et al. detail the challenges of accurate amino acid residue mapping, along with translation to transcript or gene sequence, in databases that are frequently updated. As they mention, while the number of proteins that can be lost in comparing different databases is small, the potential impact of these lost proteins can be large if one or more of the proteins is critical in the biology of the system. I found their analysis to be very thorough and I believe their findings will be highly interesting to the systems biology community. As one solution suggested by the authors, storage of the FASTA file along with the raw proteomics data would enable reproducible analysis of the proteomics data without concern over different versions of the reference data. After a thorough comparison of different databases

and their effects on proteins with different characteristics, the authors turn to cysteine and lysine residues from chemoproteomic datasets, and evaluate their reactivity vs. pathogenicity scores. Application of this analysis to G6PD highlights a couple of lysine residues in the active site of the protein, suggesting two sites for future follow-up studies. Overall this manuscript is timely, of general interest to a broad audience, and highlights the potential for using pathogenicity scores to direct genomic mutations.

We thank this reviewer for assessing our manuscript as well-written and our analyses thorough. To increase the impact of our manuscript we have added new proteomics data, new functional assay data, and a new R Shiny App, which together we believe further strengthen our manuscript, and we are hopeful that these additions will address many of the concerns highlighted by this reviewer.

Comments:

1. No reference to Figure 4D in the text.

We thank this reviewer for noticing this discrepancy. We have added a reference to this figure (now **Fig 3D**) in the text.

2. In figure 5C there are two other lysine residues that have CADD scores >25, yet neither are highlighted nor discussed. What are these residues, where do they fall on the structure, and how does this affect interpretation of the approach?

To improve the clarity and accessibility of **Fig EV4A** (formerly Figure 5C), we have completely reworked this panel and added new functional data, which together we believe more fully supports our claim that chemoproteomics data together with pathogenicity scores can be applied to identify functional residues. Our chemoproteomic reactivity dataset revealed that caspase-8 harbors two reactive cysteines (the catalytic cysteine Cys360) and a second non-catalytic cysteine (Cys409). Consistent with its function as the catalytic nucleophile, the codon of Cys360 has a high mean CADD score (29.3), whereas the codon of Cys409 has a lower CADD score (21.4), indicative that mutations that alter Cys409 should be less damaging to caspase-8. We therefore sought to test whether mutations at Cys409 would impact protein function, as indicated by the elevated measured reactivity, but not the moderate CADD score. Functional assays revealed that mutations at cys409 do indeed impact protein function, completely blocking proteolytic activity (**Fig 4E**). These data support that the reactivity measures, which indicated that this residue is likely functional, can complement the CADD data to more accurately pinpoint functional amino acids.

The original Figure 5C has been moved to the Expanded View Figures and is now **Fig EV4A**, which now includes details of the location of the additional high CADD score CpDAAs in the legend.

3. In Figure 5D, Lys171 may be directly interacting with NADP+, but lys205 appears to be outside of the direct interaction space. A different visualization may be helpful to provide a more atomic / molecular perspective on the role of these residues.

We appreciate this reviewer's careful assessment of our data. As suggested, we have provided a new perspective of G6PD active site, which includes measured distances between the lysine epsilon amines and the catalytic histidine (**Fig EV4B**, additionally shown below). We hope the reviewer will agree that this closeup clarifies the point that both K205 and K171 are within the

interacting space of the catalytic histidine. The reviewer is correct in pointing out that K205 is distal to the co-crystallized NADP+.

4. In Figure 5A and 5B, was multiple hypothesis testing used in assessing significance?

We thank the reviewer for their careful assessment of our statistical analyses. We have updated all figure legends and methods to include p. adjustment method information. From Figure 4 legend: Kruskal-Wallis nonparametric test to examine reactivity group difference, Wilcoxon test used for pairwise comparisons (BH-adjusted p-values, **p. adj* = 0.04, ***p. adj* = 0.0037, ****p. adj* = 0.00013). Median of the CADD38 max codon scores with bootstrapped 95% confidence intervals for reactive groups are: low CpD Cys 27.3 [26.9, 28.0], medium CpD Cys 28.55 [27.80, 29.05], high CpD Cys 31 [28.8, 32.0], low CpD Lys 29.5 [29.3, 29.6], medium CpD Lys 29.25 [28.85, 29.50], high CpD Lys 29.05 [28.50, 29.55].

Reviewer #3:

The manuscript integrates chemoproteomic data with predictions of genetic variant pathogenicity, to identify codons of highly reactive cysteines that are enriched for genetic variants that are predicted to be more deleterious. According to the authors this will advance prioritization of putative druggable sites. Specifically This manuscript describes variability between data found in Ensembl, GENCODE, NCBI, and CCDS. They propose using version IDs or a cross reference file from UniProtKB that displays the CCDS canonical isoform ID mapped to a stable Ensembl protein ID to map chemoproteomic data between databases. They then mapped the CpDAA coordinates to a panel of functional annotation scores and calculated the correlation of scores for overlapping codon coordinates of CpD cysteine and lysine residues. While this paper has some interesting preliminary results I don't think it is of sufficient significance to be published in a journal like MSB.

We thank this reviewer for their careful analysis of our manuscript and are delighted to note that they find some of our analyses interesting. To address this reviewer's points about the overemphasis of the data integration problem and the preliminary nature of our findings, we have taken the following steps, which are also described in more detail below: 1) Reworked the first half of the manuscript to significantly shorten the data integration section, 2) Added new analysis of the Clinvar database (**Fig 3F**), 3) Added new chemoproteomics datasets and additional meta-analyses, 4) Added functional validation studies with the cysteine protease

caspase-8, 5) Added a clear README protocol to facilitate use of our custom code, and 6) built an R Shiny App to facilitate reader visualization and post-publication analysis of our data. Collectively, we believe that these new data directly address in a compelling way Referee 3's recommendation to increase the significance of our findings to enhance general interest in our paper.

1. The most significant issue is that the major conclusion is not supported by the data. The authors write "Taken together, this analysis highlights the utility of integration of pathogenicity predictions to improve stratification of chemoproteomics data." (page 18). This is a strong claim especially since there is no experimental validation for any of the predictions. It would be critical to test some of the predictions experimentally.

To address these points and to further flesh out the second half of the manuscript we have added the following key analyses, which we believe substantially strengthen our manuscript: First, we added additional analysis of CpDAAs identified in Clinvar, which revealed that chemoproteomic detected lysines showed significant enrichment of disease-associated variants (**Fig 3F**, blue). Second, we generated additional new chemoproteomic cysteine reactivity datasets, which identified in aggregate 4017 cysteines, of which 322 were found to be hyperreactive (**Fig EV5**). When compared to the prior published dataset (Weerapana et al., Nature 2010), which we initially analyzed, our new dataset represents a significant technical advance, as it contains four-fold greater cysteines than were identified previously. This rich dataset allowed us to further verify that our exciting finding that the codons of highly reactive CpDAAs are enriched for high pathogenicity scores. Gratifyingly, our initial finding was reproduced with this new and much larger dataset (**Fig EV5A**), supporting both the validity of our approach and the robustness of our findings. Third, we conducted functional studies on the cysteine protease caspase-8, which provide additional concrete evidence that the combination of CpDAA data with predictions of pathogenicity can aid new discoveries. Our chemoproteomic reactivity dataset (**Dataset EV18** and **Fig EV5C**) revealed that caspase-8 harbors two reactive cysteines (the catalytic cysteine Cys360) and a second non-catalytic cysteine (Cys409). Consistent with its function as the catalytic nucleophile, the codon of Cys360 has a high mean CADD score (29.3), whereas the codon of Cys409 has a lower CADD score (21.4), indicative that mutations that alter Cys409 should be less damaging to caspase-8. We therefore sought to test whether mutations at Cys409 would impact protein function, as indicated by the elevated measured reactivity, but not the moderate CADD score. Functional assays revealed that mutations at Cys409 do indeed impact protein function, completely blocking proteolytic activity (**Fig 4E**). These data support that the reactivity measures, which indicated that this residue is likely functional, can complement the CADD data to more accurately pinpoint functional amino acids.

Collectively we believe that these data provide a compelling case for the utility of integration of pathogenicity predictions to improve stratification of chemoproteomics data. We believe that our functional analysis of mutation-dependent changes to caspase-8 activity provides an exciting first use case for our approach, and we hope that this reviewer would concur with us that functional validation for all identified amino acids is beyond the scope of this current study.

2. Another important issue is that the results and their significance are very thin. While the generation of the database by combining different data types should be commended, the analysis of this set is very limited. The results section focuses too much on how the data was integrated and not on what we can learn from it.

We agree with this reviewer that the data integration portion of the manuscript was cumbersome and overly long. To address this point, as noted above, We have consolidated Figures 1-3, which we believe improves the clarity and accessibility of the central message. We have also consolidated the corresponding results section for these figures for clarity. We have further added an additional table (**Table 1**) which summarizes the key actionable strategies for improved inter-datatype mapping. We have also included an additional README document available at <https://github.com/Mfpfox/MAPPING> to help users work with their data sets to accurately map genomic features. Lastly, in addition to providing all code on GitHub, the "CpDAA Database" is now housed as an R Shiny App (<https://mfpalafox.shinyapps.io/CpDAA/>), which allows readers to further visualize and explore the exciting correlations between chemoproteomic and pathogenicity prediction datasets. We hope this reviewer will concur with us that rigor and reproducibility are key to multi-omic data analysis. As such, we believe that it is essential for the field to establish guidelines and best practices to improve the reproducibility of such analyses, and we hope that this reviewer will concur that our revised manuscript helps to establish those practices, which represents a significant advance for the field.

As detailed above, we have incorporated a number of new analyses and datasets into the updated manuscript. New additions to the manuscript included analysis of Clinvar variants, generation of new proteomic data, and functional studies with the cysteine protease caspase-8. We are hopeful that this reviewer will agree with us that these additions substantially increase the impact and significance of our study.

3. There are also significant problems with the writing of the manuscript. It is hard to distinguish between the results and methods. For example, Pages 5-8 read more like Methods and make the manuscript hard to read. The results in section 3 (pages 11-14), are also very descriptive and read like methods.

We thank this reviewer for their suggestion that we should move some of the overly detailed results to the methods section. We have integrated this suggestion into our revised manuscript, which now contains a streamlined data integration section with significant portions of the sections noted by the review moved to the methods and to the dataset mapping README file, which is available on GitHub, together with all code needed to reproduce these analyses.

4. There are also serious organizational issues with the manuscript. For example, in page 7, second paragraph "Specific releases were prioritized... ". The manuscript does not describe the importance or reason for this criteria when it is first introduced. It does include some information about (1) on page 15. This delayed partial explanation makes the selection criteria unclear for a large portion of the manuscript.

We thank this reviewer for noting these organizational discrepancies. We have moved this section later in the text to improve the organizational flow of the manuscript. We have additionally added the following sentence to clarify the rationale for these criteria "Collectively, these criteria allowed us to assess data common mapping strategies and to enable downstream integration of our CpDAAs, with predictions of pathogenicity."

5. Page 8. "We assembled a comprehensive inventory of all database updates for Ensembl, GENCODE, NCBI, and CCDS (Table S4) between the dates of August 2013 - July 2019 (6 years)". This paragraph does not describe the quantitative results from this analysis, but instead states that changes accumulate causing data loss.

This section has been reworded to the following: “Accurate residue-level mapping between sequences from different database sources is further complicated by the frequent and unsynchronized update cycles of independent databases (**Fig 1B, Dataset EV2**). Quantification of the average update cycle for each database across this time period revealed that UniProtKB has the shortest mean update cycle (~6 weeks) (**Fig 1C**). In contrast, NCBI is only updated yearly. These different update cycles can create a lag between versions of databases used to create identifier cross-reference (a.k.a. xref) files (**Appendix Table S1**). For example, stable ID mapping files provided by Ensembl for UniProtKB proteins may not share identical sequences if not used within the short 4-week window between UniProtKB updates. These changes can accumulate over time and can lead to data loss and, worse, residue mis-annotation”

6. Page 11. *"ID cross-reference files (Table S2) that are released by Ensembl and UniProtKB ". What is the rationale for selecting G6PD for investigation? While there is a small description of clinical manifestations of G6PD deficiency on page 17, much like the comments about page 7 the placement seems delayed.*

We have substantially reworked this section of the manuscript and we hope this reviewer will agree that the new version addresses many of the issues identified here. G6PD was selected as a representative protein due both to the multiple reactive residues featured within its sequence and to its established genetic variants that are linked to both rare and common disorders. However, we concur with both reviewer 3 and reviewer 2 that the perceived impact of the G6PD example was relatively modest. As such, we have both modified the G6PD figure to more clearly show the active site and have additionally moved the figure to the Expanded View Figures (Now **Fig EV4**). The new **Fig 4D** shows the average CADD scores mapped to the X-ray structure of caspase-8 bound to a peptide inhibitor. As a result of our compelling new proteomic and functional data (detailed above), we believe that the rationale for showcasing caspase-8 is much more compelling than was the case of G6PD.

7. Page 12. *The manuscript recommend using version IDs and then say why they are too challenging to use in the same paragraph*

We agree with this reviewer that this section of the manuscript was unnecessarily confusing. The paragraph discussion the use of versioned vs stable identifiers that the reviewer is referring to has now been removed from the text.

8. Page 13. *The following statement is confusing. "To assess how pervasive multi-mapping is across the whole CpDAA dataset, we quantified the mean number of Ensembl IDs per UniProtKB IDs. We found that the mean number of Ensembl IDs per UniProtKB IDs was significantly increased in versioned IDs compared to stable IDs, both for Ensembl gene and transcript IDs ". Isn't that redundant- by having version IDs you increase the number of IDs?*

We have modified this section to the following, which we believe better highlights the the distinctions between stable and versioned IDs and the rationale behind this analysis:

“To assess how pervasive multi-mapping is across the entire CpDAA dataset, we quantified the mean number of Ensembl IDs per UniProtKB ID. We counted both versioned and stable Ensembl IDs types (gene, transcript, and protein IDs), for all CpD UniProtKB proteins grouped by single (**Fig EV1C**) or multi-isoform (**Fig EV1D, Dataset EV11**) associated stable IDs. We suspected that database updates for all data types (gene, transcript, and protein) and the presence of UniProtKB isoforms would contribute to the observed multi-mapping of CpD protein

IDs in our dataset. Of note, Ensembl versioned IDs indicate changes to the associated sequence rather than the presence of isoforms. For example, for protein tropomyosin alpha-4 chain (TPM4, P67936), during the update from v96 to v97, the stable protein identifier showed version change from '.3' to '.4' (ENSP00000300933.3 to ENSP00000300933.4), which corresponds to a difference of 165 amino acids in the primary sequence caused by the update (**Dataset EV10**). Not surprisingly, we found that UniProtKB stable identifiers with multiple associated protein isoforms have on average a higher average of cross-referenced Ensembl ID types per UniProtKB stable identifier compared to UniProtKB stable IDs associated with only one protein isoform. In addition, single isoform UniProtKB stable IDs are more likely to cross-reference identical Ensembl proteins compared to multi-isoform UniProtKB stable IDs (**Appendix Fig S3 and S4**)."

9. Page 13 last paragraph. "We identified the top stable IDs ...". This analysis identified 49 UniProtKB IDs, is this a significant number?

We agree with this reviewer that 49 IDs represents a seemingly small fraction of the total data. However, if a key protein of interest for a particular study is lost due to mismapping, we believe that it would constitute a significant data loss. Acknowledging that the phrasing may be perceived as misleading, we have reworked this section of the manuscript and have moved the following statement to the legend of Appendix Fig S5 "Source data is shown in **Source Data for S5 Table**, which includes the 49 UniProtKB IDs that had no canonical sequence equivalent in all five Ensembl releases analyzed and CpDAA index differences for most detected cysteine or lysine positions."

10. Page 15 second paragraph. "When comparing only machine learning-based scores for substitutions ..." Only place the term machine learning is used in the manuscript. What do you mean by machine learning here? It doesn't feel like it fits in the document.

We agree with this reviewer that the mention of machine learning-based scores does not add to the manuscript. To address this point, we have modified the text to the following: "For the subset of scores that provide deleteriousness scores for all possible non-synonymous variants, we did not observe substantial differences between the correlation of scores for detected and undetected CpD lysines or CpD cysteines (**Appendix Fig S7, Dataset EV15**)."

Thank you again for sending us your revised manuscript. We have now heard back from the two reviewers who were asked to evaluate your study. As you will see below, reviewer #1 is supportive of publication. However, reviewer #3 still raises several concerns, the most prominent being the absence of experimental validations of the predictions for previously unknown functionally important reactive cysteines or lysines. We think that the proof-of-principle validation analyses of caspase-8 would seem sufficient at this point and given that reviewer #1 does not seem to be concerned about the lack of further validations we think that further experimental validations are not mandatory for the acceptance of this work. If you have such analyses at hand and would like to include them, this would of course be a welcome addition, but we leave the decision up to you. In sum, I would like to invite you to perform a minor revision, addressing the remaining more minor concerns of reviewer #3.

Moreover, we would ask you to address a few remaining editorial issues listed below.

REFEREE REPORTS

Reviewer #1:

In this revised version of the manuscript the authors have made significant improvements to the original submission, improving clarity (condensing the first three figures), utility (deployment of a new app), as well as providing a useful set of guidelines (Table 1), and also providing novel proof-of-concept data to emphasize the potential of their approach to lead to new findings.

The authors have addressed all my concerns, and hopefully have found the review process to be useful! I happily recommend this work for publication.

Reviewer #3:

In this revised version the authors significantly improved the writing, and added some new analyses. For example, they find that codons of highly reactive CpDAAs are enriched for high pathogenicity scores, which is interesting.

However, I still feel that this manuscript is very technical and provides a resource whose biological relevance needs to be demonstrated. The examples given are more anecdotal (e.g. caspase-8), and I feel that predictions for previously unknown functionally important reactive cysteines or lysines should be tested experimentally for a journal like MSB. While the new R-Shiny application is nice, I don't think it really adds to the significance of the work. Here are some specific comments:

The Result section still includes many technical details that are only marginally interesting. For example, mapping the identifiers on page 6-7 of the Results is very detailed and hard to read. I don't see the point of it. Section 2 of the results (page 8-9) presents some findings about mapping of the sequences and then discusses some differences between Uniprot 2012 and 2018 which is not that exciting. It is still not clear to me why the authors validate with the 2012 UniProtKB. I understand they compare it to the 2018 UniProtKB later, but why did initially select this version for the comparison? It is good that they evaluated different methods for inter-database mapping but I felt that they could have made a stronger case for why they actually went with the "stable" identifiers rather than the other options they examined.

If cysteine alone does not show significant enrichment does adding them into the model help or is it just the case of adding more variables into a multivariable model causing marginally better returns by virtue of the fact that there are just more descriptors.

A Point-by-Point response to all reviewer comments:**Reviewer #1:**

In this revised version of the manuscript the authors have made significant improvements to the original submission, improving clarity (condensing the first three figures), utility (deployment of a new app), as well as providing a useful set of guidelines (Table 1), and also providing novel proof-of-concept data to emphasize the potential of their approach to lead to new findings. The authors have addressed all my concerns, and hopefully have found the review process to be useful! I happily recommend this work for publication.

We thank this reviewer for their positive assessment of our revised manuscript.

Reviewer #3:

In this revised version the authors significantly improved the writing, and added some new analyses. For example, they find that codons of highly reactive CpDAAs are enriched for high pathogenicity scores, which is interesting. However, I still feel that this manuscript is very technical and provides a resource whose biological relevance needs to be demonstrated. The examples given are more anecdotal (e.g. caspase-8), and I feel that predictions for previously unknown functionally important reactive cysteines or lysines should be tested experimentally for a journal like MSB. While the new R-Shiny application is nice, I don't think it really adds to the significance of the work. Here are some specific comments:

We are delighted that this reviewer found our manuscript writing improved, and we thank them for their careful analysis of our manuscript. To improve the accessibility of our manuscript to a wider audience, we have revised several sections, described in more detail below.

Looking beyond the previously uncharacterized cysteine in caspase-8 that we identified for the first time as functionally important for enzymatic activity, we concur with this reviewer that understanding whether additional reactive cysteines are functionally significant is an important area of research that merits further study. In fact, a substantial motivation for this current study was to provide a new approach to stratify the likely functional significance of chemoproteomic detected cysteines in a high throughput manner. Given the challenges associated with the high-throughput and global analysis of amino acid function (detailed both by our study and prior manuscripts), we hope that this reviewer will agree that the truly comprehensive analysis of cysteine functionality (hundreds to thousands of residues) that is required to make statistically rigorous assessments of functional significance is beyond the scope of the current study. We are enthusiastic about the possibility of reporting such information in future studies.

1. *The Result section still includes many technical details that are only marginally interesting. For example, mapping the identifiers on page 6-7 of the Results is very detailed and hard to read. I don't see the point of it.*

While we do believe that there is some value in our detailed assessment of data integration strategies, both for enhancing rigor and reproducibility of multi-omic studies and for establishing norms and guidelines for the community, in response to this reviewer comment we have opted to streamline the manuscript and have substantially reworked several sections to improve the accessibility our our manuscript to a wide audience.

A large portion of the results on page 6 paragraph 1 have been condensed to a single sentence outcome of aggregating publicly available chemoproteomics data:

“For this we aggregated publicly available cysteine and lysine chemoproteomics datasets (Backus et al. 2016; Hacker et al. 2017; Weerapana et al. 2010), resulting in a total of 6,510 CpD cysteines and 9,327 CpD lysines detected in 4,119 unique proteins.”

Additionally, we revised the bulk of the text in page 6 paragraph 2 and merged the remaining sentences into two sentences, which we believe provide simpler and more accessible description of our approach:

“As our overarching objective was to characterize CpDAAs using functional annotations based on different versions of protein, transcript, and DNA sequences (Fig 1A), our next step was to develop a high-fidelity data analysis pipeline for intra- and inter-database mapping. To guide our analyses, we first referenced established methods for such data mapping, including ID mapping (Meyer, Geske, and Yu 2016; Huang et al. 2008; Xin et al. 2016), residue-residue mapping (David and Yip 2008; Martin 2005; Dana et al. 2019), and residue-codon mapping (Li et al. 2016; Zhou et al. 2015) (See Appendix Table S1 for detailed descriptions of each type of mapping).”

2. Section 2 of the results (page 8-9) presents some findings about mapping of the sequences and then discusses some differences between Uniprot 2012 and 2018 which is not that exciting. It is still not clear to me why the authors validate with the 2012 UniProtKB. I understand they compare it to the 2018 UniProtKB later, but why did they initially select this version for the comparison?

To clarify why our data was first validated against the 2012 UniProtKB reference file, the published proteomics data that we reanalyzed here were originally processed using a UniProtKB reference file from November 2012. To achieve high fidelity data mapping, we first opted to verify that the data matched with an author-provided 2012 UniprotKB reference file (no longer publicly available due to UniProtKB archiving policies). This first remapping step confirmed, gratifyingly, that >99% of the identifiers indeed mapped to the 2012 reference. The remaining 1% can likely be rationalized by idiosyncrasies in the original data processing workflow, such as slight differences between the database searched and that provided by the authors.

We have substantially reworked this section of the manuscript and we hope that the revised version is clearer and more accessible to the reader:

“Therefore, we next assessed whether and to what extent updates to the canonical sequences assigned to UniProtKB stable identifiers resulted in mismapping. To confirm the integrity of our CpDAA dataset, we started this process by validating that over 99% of the CpDAA protein IDs and residue positions matched with those found in a 2012 UniProt FASTA file, corresponding to the reference proteome originally used to process the datasets (see methods and Dataset EV1). The small fraction of data lost was due to missing stable identifiers and mis-matched CpDAA positions, which likely stems from slight inconsistencies between the original processing pipeline and our current workflow.”

3. *It is good that they evaluated different methods for inter-database mapping but I felt that they could have made a stronger case for why they actually went with the "stable" identifiers rather than the other options they examined.*

We appreciate that this reviewer valued our assessment of multiple data mapping strategies, as we believe that these represent important variables in large scale data integration strategies.

To clarify our decision to use stable identifiers, we have added the following text to page 8 paragraph 1:

*“Proteomics datasets, including the published CpDAA datasets, are routinely searched against FASTA files containing only canonical UniProtKB proteins (**Appendix Table S1**), for two main reasons. First, canonical proteins reduce the redundancy and complexity of proteome search databases. Second, these sequences are identified by stable identifiers (also known as the UniProtKB primary accessions) and offer the seeming advantage of remaining constant through database update cycles. However, one particularly confusing aspect of the stable identifier is that the word “stable” in this context does not mean permanent or immutable. Specifically, the associated sequence linked to a stable identifier can change over database releases. These changes can cause residue mismapping, where the amino acid position is correct in one release but incorrect in a future updated database release.”*

We concur with this reviewer that there are significant limitations to stable identifiers, as described above. To address these limitations, we developed a multi-step data mapping pipeline to match the residue numbers with equivalent genomic-level information provided by the ClinVar, CADD v1.4 and dbNSFPv4 databases, as detailed in the text and shown schematically in Figure EV2.

We hope that this reviewer will agree that the above text clarifies the importance of the critical mapping step, as well as the potential limitations.

4. *If cysteine alone does not show significant enrichment does adding them into the model help or is it just the case of adding more variables into a multivariable model causing marginally better returns by virtue of the fact that there are just more descriptors.*

We agree with this reviewer that the addition of more variables does not necessarily increase the accuracy of multivariable models, such as predictions of pathogenicity. Our approach does not develop multivariable modeling for pathogenicity prediction but rather explores whether specific features for reactivity are associated with pathogenicity in human disease. Our data identified an enrichment for high predictions of pathogenicity for the subset of CpD cysteines that are highly reactive (and for all CpD lysines). This suggests that highly reactive residues may be informative for future algorithms, but these need to be further explored.

Thank you again for sending us your revised manuscript. We are now satisfied with the modifications made and I am pleased to inform you that your paper has been accepted for publication.

Corresponding Author Name: Backus, Arboleda

Manuscript Number: MSB-20-9840